

# Mature riparian alder forest acts as a strong and consistent carbon sink

Alisa Krasnova[1,2], Kaido Soosaar[1], Svyatoslav Rogozin[1], Dmitrii Krasnov[3], Ülo Mander[1]

[1]Department of Geography, Institute of Ecology and Earth Sciences, University of Tartu, Tartu, 50090, Estonia
[2]Department of Forest Ecology and Management, Swedish University of Agricultural Sciences, Umeå, 90183, Sweden
[3]Institute of Forestry and Engineering, Estonian Life Science University, Tartu, 51006, Estonia

*Correspondence to*: Alisa Krasnova (alisa.krasnova@ut.ee)

**Abstract.** Alder forests are widely spread across Northern Hemisphere, frequently occupying riparian buffer zones and playing a key role in enhancing soil fertility through symbiosis with nitrogen-fixing bacteria. Despite their ecological significance, studies on carbon (C) and water ($H_2O$) exchange in alder forests remain scarce, particularly in the context of hydroclimatic variability and extreme weather events. In this study, we used eddy-covariance flux measurements from three contrasting years to assess the C balance and $H_2O$ exchange of a mature riparian grey alder forest in the hemiboreal zone in Estonia. The site was a strong and consistent carbon sink with annual net ecosystem exchange (NEE) ranging from -496 to -663 g C m$^{-2}$ y$^{-1}$, gross primary production (GPP) from -1258 to -1420 g C m$^{-2}$ y$^{-1}$ and ecosystem respiration (ER) from 595 to 923 g C m$^{-2}$ y$^{-1}$. Evapotranspiration (ET) varied from 194 to 342 kg $H_2O$ m$^{-2}$ y$^{-1}$ and ecosystem water use efficiency (EWUE) was 4.2 - 6.5 g C kg $H_2O^{-1}$. The drought and heatwave year (2018) featured the highest net carbon uptake, driven by an increase in GPP during spring and a reduction in ER during late summer and autumn. A minor impact of drought on GPP combined with a 35% reduction in ET in 2018 lead to peak values of EWUE in response to $H_2O$ limitation. In 2019, we found no evidence of a short-term drought legacy effect, as carbon exchange components recovered to the 2017 levels and ET was the highest out of years. Given that this forest is beyond the typical harvestable age, its strong and consistent carbon sequestration, combined with high short-term resilience, provides valuable insights for sustainable forest management. These findings highlight the potential of riparian grey alder forests to maintain productivity under hydroclimatic variability, reinforcing their role in regional carbon cycling as a part of natural climate mitigation solutions.

## 1. Introduction

Terrestrial ecosystems play an essential role in mitigating the rise of atmospheric carbon dioxide ($CO_2$) concentrations and restraining global warming (Pan et al., 2011; Piao et al., 2020). Over the preceding decades, they have effectively sequestered approximately one-third of the total industrial carbon (C) emissions (Friedlingstein et al., 2022). Forest ecosystems, in particular, typically act as net C sinks, with rate of photosynthetic uptake surpassing respiratory emissions on the annual scale (Harris et al., 2021). However, the strength of this C sink is contingent upon various factors, including forest age, tree species composition, climatic conditions, soil properties, and management practices (Winkler et al., 2023). Moreover, under certain conditions, a local C-sequestering forest may transition to a state of C neutrality or even become a net C source, thereby affecting ecosystem-atmosphere interactions at a regional scale (Hadden and Grelle, 2016; Lindroth et al., 1998). Therefore, it is critical to evaluate the sustainability of a local forest climate mitigation potential in the face of varying climatic events (Allen et al., 2010; Bonan, 2008; Teskey et al., 2015).

Water availability is one of the crucial factors in forest survival, and droughts could be one of the key reasons for forest mortality (Allen et al., 2010; Breshears et al., 2005; Cavin et al., 2013; Haberstroh et al., 2022; McDowell et al., 2008). The frequency and severity of extreme climate events, including droughts, have been growing in recent decades and are expected to continue in the future (Fischer et al., 2021; Trenberth et al., 2014). In 2018, Europe faced a drought that was considered the



most severe in the last 250 years (Gutierrez Lopez et al., 2021; Hari et al., 2020), causing reduced C uptake and elevated tree

mortality rates (Bastos et al., 2020; Buras et al., 2020; Haberstroh et al., 2022; Senf and Seidl, 2021; Smith et al., 2020). Thus,

it is essential to quantify the C uptake capacities of different forests both during and following drought conditions to better

understand their resilience and inform management strategies for enhancing forest sustainability.

Grey alder (Alnus incana (L.) Moench.) is a typical pioneer species frequently occupying riparian zones and is widely spread

in North America and Europe (Caudullo et al., 2017). Alder plantations can mitigate C losses in rewetted peatlands (Huth et

al., 2018) and improve the soil structure of skid trails (Warlo et al., 2019). Their high adaptability also makes alders suitable

for afforestation of post-industrial sites (Krzaklewski et al., 2012). Owing to their symbiosis with atmospheric nitrogen-fixing

bacteria (Benson, 1982; Rytter et al., 1989), alder trees play an essential role in forest soil nitrogen enrichment  (Mander et al.,

2008, 2021; Soosaar et al., 2011). Moreover, due to their rapid growth, alder species are frequently chosen for riparian buffer

zones and short-rotation forestry (Aosaar et al., 2012; Rytter and Rytter, 2016; Uri et al., 2017). However, there are surprisingly

few studies on the C uptake potential and water use efficiency of alder forests, particularly in the context of extreme weather

events.

The net ecosystem production (NEP) of a grey alder forest chronosequence in Estonia was previously estimated by Uri at al.

(2017), utilising the traditional C budgeting method. Their findings indicated that while most grey alder stands functioned as

C sinks, a young (9-year-old) and a mature (40-year-old) site acted as moderate C sources. This was attributed to elevated

heterotrophic respiration at both sites and, in the case of the young stand, low net primary production. However, the authors

noted that interannual variations in NEP were primarily driven by climatic factors rather than stand age. These findings

highlight the need for further assessments of net C uptake under varying weather conditions, particularly in mature grey alder

forests.

In this study, we aim to investigate the C and water exchange of a mature riparian alder forest stand in the hemiboreal zone.

While our previous research (Krasnova et al., 2022) examined how several forested ecosystems in Estonia responded to

elevated temperatures during the 2018 heatwave, water fluxes were not analysed. Moreover, the effects of stress factors on

forest ecosystems can become more pronounced in the years following exposure (Anderegg et al., 2015; Kannenberg et al.,

2020), which was beyond the scope of our previous study. Therefore, the specific objectives of this study are to (1) quantify

the C and water exchange of an alder forest under varying hydroclimatic conditions; (2) investigate the influence of different

soil moisture regimes on alder forest C exchange and water use efficiency; (3) evaluate the presence of legacy effects and the

long-term sustainability of grey alder forests as a nature-based solution for climate mitigation.

## 2. Methods

### 2.1 Study site and footprint area

The ecosystem in our study is a mature 40-year-old riparian grey alder (*Alnus incana* (L.) Moench) forest stand located on a

former agricultural land in southern Estonia. The terrain is flat, formed at the bottom of former periglacial lake systems, with

an average elevation of 32 m a.s.l. and a 1% inclination slope towards a tributary of the Kalli River. The average annual air

temperature is 5.8 °C, whereas in July and January, the mean air temperatures are 17.0 °C and -6.7 °C, respectively (Kupper

et al., 2011). The soil is Gleyic Luvisol, with a 15-20cm humus layer.  The top 10 cm soil C and N content were 3.8% and

0.33% (Mander et al., 2022), resulting in the C:N ratio of 11.5.



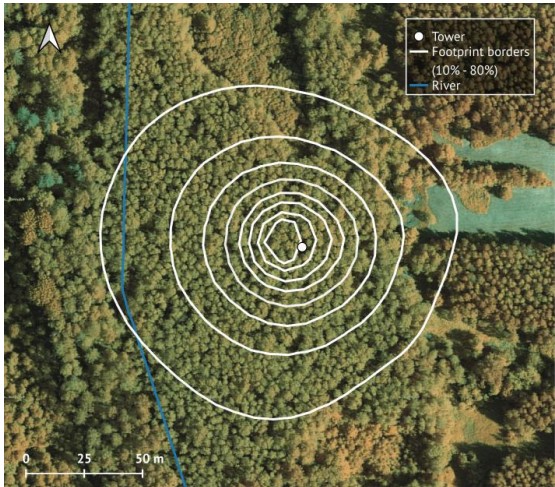

**Figure 1: Tower footprint area (10%-80%), Kljun model (Kljun et al., 2015); the blue line indicates the location Kalli River. Map data: Estonian Land Board (Maa-amet), accessed via QGIS.**

The total footprint area (Fig. 1) is 1.65 ha, 85% of which (1.41 ha), is covered by grey alder. The average stand height is 17.5

m, the stand density is 1520 trees per ha, the mean stem diameter at breast height is 15.6 cm, and the basal area 30.5 $m^2$ $ha^{-1}$ (Mander et al., 2022). The understory is dominated by herbs (*Filipendula ulmaria* (L.) Maxim., *Aegopodium podagraria* L., *Cirsium oleraceum* (L.) Scop., *Geum rivale* L., *Crepis paludosa* L., mosses (*Climacium dendroides* (Hedw.) F. Weber & D. Mohr, *Plagiomnium* spp. and *Rhytidiadelphus triquetrus* (Hedw.) Warnst. Moench,), shrubs (*Rubus idaeus* L., *Frangula alnus* L., *Daphne mezereum* L.) and young trees (*Alnus incana*, *Prunus padus* L.).

**2.2 Instrumentation**

The eddy-covariance setup consisted of a fast 3-D sonic anemometer Gill HS-50 (Gill Instruments Ltd., Lymington, Hampshire, UK) and enclosed $CO_2$ and $H_2O$ gas-analyser LI-7200 (LI-COR Biosciences, Lincoln, NE, USA) measuring with 10Hz frequency. The instruments were mounted on top of a 21m scaffolding tower in spring 2017; the first measurements started on the 15th of May 2017. Air temperature and humidity (Rotronic HC2A-S3; Rotronic AG, Bassersdorf, Switzerland)

and photosynthetically active radiation (PAR; LI-190SL; LI-COR Biosciences, Lincoln, NE, USA) were measured in a tower at 5 m height for air temperature and relative humidity and 25 m height for PAR (above the forest canopy). Soil temperature (107, Campbell Scientific Inc., Logan, Utah, USA) and soil water content (ML3 ThetaProbe, Delta-T Devices, Burwell, Cambridge, UK) sensors were installed at 10 cm depth in the end of July 2017. WTD was measured manually in groundwater wells next to the soil chambers on each sampling day.

**2.3 Fluxes calculation and post-processing**

The fluxes of $CO_2$ and latent heat (LE) were calculated as a covariance between vertical wind speeds and $CO_2$ (or $H_2O$) concentrations using EddyPro software (version 6.3.0, LI-COR Biosciences, USA) and averaged over the 30-minute intervals. In the absence of a storage measuring profile system, we estimated flux storage using the tower-top method, which utilised half-hourly $CO_2$ concentration measurements from the EC system. Net ecosystem exchange (NEE) was then calculated as the

sum of eddy flux and storage. To eliminate periods of underdeveloped turbulence, we applied friction velocity filtering; the threshold of 0.28 m $s^{-1}$ for 2017-2018 and 0.22 m $s^{-1}$ for 2019 were calculated with a moving point test (Papale et al., 2006). Fluxes during the half-hours with friction velocity values below these thresholds were removed from the analysis. To ensure adequate mixing conditions throughout the measurement period, we opted to remove not only nighttime half-hours but also daytime NEE values associated with low friction velocity estimates.



A previous study conducted at the same site by Mander et al. (2022) considered unaccounted advection as a possible reason for the discrepancy between soil and ecosystem scale fluxes. To identify the periods when advection was significant, we applied the filtering method following Wharton et al. (2009) and Chi et al. (2019). Turbulence intensity parameters ($I_w$ and $I_u$) were calculated for each half-hour as the ratios of vertical and horizontal wind velocity to turbulence intensity, respectively. For any half-hour, if $I_w$ or $I_u$ was outside of the window of mean plus one standard deviation estimated for the entire

measurement period, advective conditions during this half-hour were considered non-negligible, and NEE and LE were filtered out.

The remaining spikes in the dataset could be attributed to the simplification of the flux storage calculation procedure or the instrumental failure. Therefore, fluxes outside the common range (mean ± 3×standard deviation) were filtered out over a 14-day moving window (151 half-hour values). Overall, the final quality-controlled values were 60% in 2017, 66% in 2018 and

65% in 2019. Evapotranspiration (ET) was then calculated by dividing the filtered LE by the latent heat of vaporisation (Allen et al., 1998).

In order to obtain fluxes aggregated over various time scales, we gap-filled NEE and ET using XGBoost as recommended by Vekuri et al. (2023). The hyperparameters were tuned during 5-fold cross-validation and included maximum tree depths (3, 5, 10, 15), regularization strength with default 0, data sampling ratios (0.5, 0.75, 1), feature sampling ratios (0.4, 0.6, 0.8, 1), and

minimum child weights (2, 5, 10). The hyperparameters were determined using all available data. A squared loss with a default learning rate of 0.1 was used as an objective function.

The partitioning of NEE into gross primary production (GPP) and ecosystem respiration (ER) was performed with the "nighttime" method utilising the ReddyProcWeb tool (Wutzler et al., 2018). Nighttime respiration was considered equal to nighttime gap-filled NEE values, while daytime ER was modelled in ReddyProcWeb using the air temperature dependence of

measured nighttime values. GPP was then calculated as a difference between gap-filled NEE and modelled daytime ER. Following the micrometeorological convention, negative flux denotes uptake, while positive flux is a release from the ecosystem into the atmosphere.

**2.4 Additional parameters and statistical analysis**

To quantify forest resistance and resilience, we estimated two additional parameters: ecosystem water use efficiency (EWUE)

as indicator of a forest's adaptability to changes in water resources (Huang et al., 2015; Keenan et al., 2013; Yang et al., 2016), and canopy photosynthetic capacity ($GPP_{sat}$) as a measure of the ecosystem's functional stability (Chen et al., 2023; Musavi et al., 2017). EWUE was calculated as a ratio of GPP to ET.

To obtain $GPP_{sat}$, we used a modified version of the rectangular hyperbolic light response curve (Eq. 1), that was fitted to daytime ($Rg > 15$ W m$^{-2}$) GPP and global radiation data using a 5-day moving window. We chose this simplified light response

curve equation over the non-rectangular version used by Musavi et al. (2017) and Chen et al. (2023) because it demonstrated considerably better performance (a higher number of successful fits) with our dataset.

$$GPP = \frac{\alpha GPP_{max} R_g}{\alpha R_g + GPP_{max}},$$ (1)

where $\alpha$ (µmol J$^{-1}$) is the canopy light utilisation efficiency; Rg (W m$^{-2}$) is global radiation; $GPP_{max}$ (µmol m$^{-2}$ s$^{-1}$) is the maximum GPP, $ER_{day}$ (µmol m$^{-2}$ s$^{-1}$) is the average daytime ecosystem respiration.

We then computed $GPP_{sat}$ as GPP at Rg of 1000 W m$^{-2}$ and assigned it to the middle of each moving window, as $GPP_{max}$ does not always reflect the light saturation value. Only the values from windows with significant fit parameters ($p<0.05$) and $R^2>0.5$ were retained. Annual $GPP_{sat}$ was then calculated as the 95$^{th}$ percentile of each year's filtered values.

The start and end of each growing season (GS) were estimated using a double-logistic curve fitting method applied to daily GPP sums (Gonsamo et al., 2013). Partial correlation analysis using Spearman's coefficient ($r_s$) was performed to identify the

strongest correlation between the fluxes and the primary environmental drivers for each GS. The significance of the difference between GSs was estimated with the Wilcoxon signed-rank test using daily values and matching days of the three growing





seasons. Each GS was compared to the other two, and a Bonferroni adjustment was applied to the p-values to correct for multiple comparisons.

Light response curves (Eq. 2) were used for each GS to characterise the impact of global radiation on the daytime net C
exchange; the Lloyd and Taylor equation (Eq. 3) was used to assess the ecosystem respiration temperature response. In both cases, we utilised only measured quality-controlled NEE values. All data analysis was performed in MATLAB (2020a-2022b, Mathworks Inc.).

$$NEE_{day} = GPP + ER_{day} = \frac{\alpha GPP_{max} R_g}{\alpha R_g + GPP_{max}} + ER_{day}, \tag{2}$$

$$ER = ER_{ref} e^{E_0 \left( \frac{1}{T_{ref} - T_0} - \frac{1}{T - T_0} \right)}, \tag{3}$$

where $R_{ref}$ (μmol m$^{-2}$ s$^{-1}$) is the respiration at the reference temperature; E0 (kJ mol$^{-1}$) is the activation energy; T (°C) is the measured air temperature. $T_{ref}$ was set to 15 °C, and $T_0$ was kept constant at -46.02 °C following Lloyd and Taylor (1994).

## 3. Results

### 3.1 Meteorological conditions and growing season length

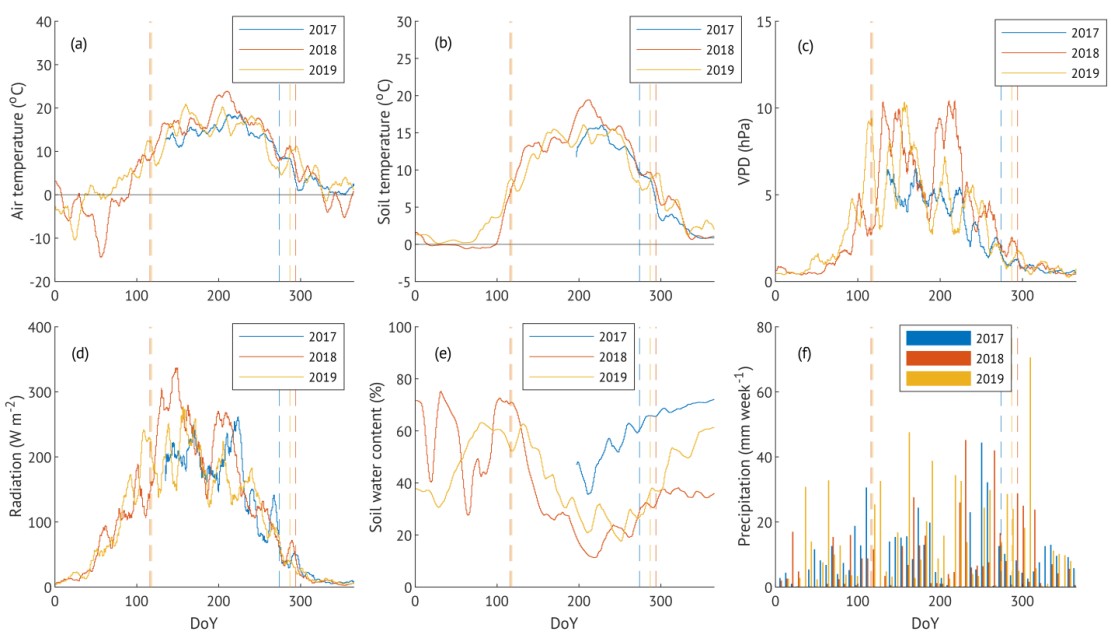

**Figure 2: Climatic conditions during the three studied years: 10-day running means of meteorological parameters (a-e) and weekly sums of precipitation (f). Dashed lines denote the beginning and end of the corresponding growing seasons.**

The meteorological conditions of the three studied years (2017-2019) were typical for the region, with below-zero air temperatures (Ta), reduced vapour pressure deficit (VPD), and low solar radiation during the winter months, and positive air temperatures, higher VPD and increased solar radiation in summer (Fig. 2). The average Ta was similar in 2017 and 2019,
whereas the 2018 GS Ta was 1.5 °C higher than the average of the other two years. Likewise, VPD was the highest in GS 2018, while the amount of rain and soil water content were the lowest. Air temperature reached the maximum in July; global radiation and VPD demonstrated two distinctive peaks in the beginning and middle of a GS. VPD peaks were absent in 2017; generally, VPD was lower and exhibited less variability in GS 2017 compared to the subsequent GSs.





Soil water content (SWC) measurements commenced in the second part of July 2017. Data from a nearby station (Appendix
Fig. 1) and visual assessment during the installation of the instrumentation confirm increased levels of SWC (standing water)
also in late spring to early summer of 2017. SWC exhibited similar patterns in 2018 and 2019, characterized by a rapid decrease
at the start of the growing season, reaching the minimum around the beginning of August 2018 (DoY 220) and the end of July
2019 (DoY 209). In both years, this dry period matched the second peaks of solar radiation and VPD, with values in 2018
higher than those in 2019. The end of July was also the driest for 2017, but with still higher SWC levels, lower radiation, air
temperature and VPD.

The growing season (GS) length was 179 and 170 days in 2018 and 2019, respectively (the flux measurements of 2017 missed
the beginning of the growing season). The start of GS was around a similar time, April 26th and April 28th, in 2018 and 2019,
but the end of GS was later in 2018 (October 21st) than in 2019 (October 14th) and 2017 (October 1st) (Fig. 2).

### 3.2 Annual and growing season accumulated fluxes

The riparian alder forest in our study acted as a strong net C sink in all three calendar years (Table 1), with total net C uptake
in 2018 being 11% higher than in the previous year and 34 % higher than NEE in the subsequent year. The difference in C
uptake between the 2018 and 2019 was primarily driven by ER, which was 36% lower in 2018, while GPP was only 11%
lower. Accumulated NEE in the active vegetation season (May-September) accounted for 97% of the total annual flux in 2018
and 95% in 2019. Using an average of 96%, we estimated the total annual NEE for 2017 to be -599.6 g C $m^{-2}$ $y^{-1}$, reflecting a
smaller C uptake than in 2018, but higher than in 2019. In a similar manner, we obtained estimates for annual GPP in 2017
(May-September GPP was on average 96% of the total), and then estimated accumulated ER as a difference between NEE and
GPP. The average annual NEE, GPP and ER over the three years of our study was -586.3 ± 84.5, -1329.8 ± 82.4 and 742.9 ±
166.3 g C $m^{-2}$ $y^{-1}$, respectively. The net C uptake in 2018 was 21% higher than the average of the other two years, while GPP
and ER were 8% and 27% lower.

Total evapotranspiration (ET) in 2018 was almost twice lower (43%) than the 2019 value. Consequently, the annual EWUE
2018 was 55% higher than in the following year. The majority of the total ET occurred during May-September (94% in 2018
and 91% in 2019), allowing for an estimate of total ET and EWUE in 2017 (Table 1). The average annual ET over the three
years was 263.6 ± 74.5 mm $y^{-1}$ and annual EWUE averaged to 5.27 ± 1.16 g C kg $H_2O^{-1}$; annual ET and EWUE in 2018 were
35% lower and 40% higher than the average of the two other years, respectively.

195 **Table 1. Average meteorological parameters (mean and standard deviation) and aggregated fluxes over the measurement years and active vegetation season (May to September)**

| | Ta | Ts | Rg | VPD | SWC | Prec. | NEE | GPP | ER | ET | EWUE |
|---|---|---|---|---|---|---|---|---|---|---|---|
| | °C | °C | $W\,m^{-2}$ | hPa | $m^3\,m^{-3}$ | $mm\,period^{-1}$ | $g\ C\ m^{-2}\,period^{-1}$ | $g\ C\ m^{-2}\,period^{-1}$ | $g\ C\ m^{-2}\,period^{-1}$ | $mm\,period^{-1}$ | $g\ C\ kg\ H_2O^{-1}$ |
| **Calendar year** | | | | | | | | | | | |
| 2017 | | | | | | | -599.6 | -1310.7 | 711.1 | 255.1 | 5.1 |
| 2018 | 7.6 ± 10.6 | 7.5 ± 6.8 | 113.7 ± 200.8 | 3.4 ± 4.9 | 0.39 ± 0.19 | 452.6 | -663.4 | -1258.2 | 594.9 | 193.7 | 6.5 |
| 2019 | 7.7 ± 8.5 | 7.4 ± 5.7 | 102.5 ± 182.6 | 3.1 ± 4.3 | 0.43 ± 0.14 | 710.0 | -495.9 | -1419.8 | 922.8 | 342.0 | 4.2 |
| **May-September** | | | | | | | | | | | |
| 2017 | 14.5 ± 4.2 | 13.8 ± 1.8* | 173.5 ± 229.4 | 4.2 ± 3.5 | 0.52 ± 0.09* | 239.8 | -575.6 | -1258.3 | 682.7 | 236.0 | 5.3 |
| 2018 | 16.9 ± 5.6 | 14.6 ± 2.8 | 204.2 ± 253.7 | 6.5 ± 6.1 | 0.28 ± 0.14 | 241.4 | -775.5 | -1215.0 | 439.5 | 182.3 | 6.7 |
| 2019 | 14.8 ± 5.8 | 13.1 ± 2.7 | 167.8 ± 220.4 | 5.0 ± 5.0 | 0.36 ± 0.14 | 376.4 | -634.9 | -1351.4 | 715.4 | 311.7 | 4.3 |

* starting from 24.07.2017

Total GPP in the active vegetation season of 2018 was only 3% lower than the previous year but 10% lower than the following
year. Total ER over May-September varied between the years, with the lowest aggregated C release in 2018 (39% lower than





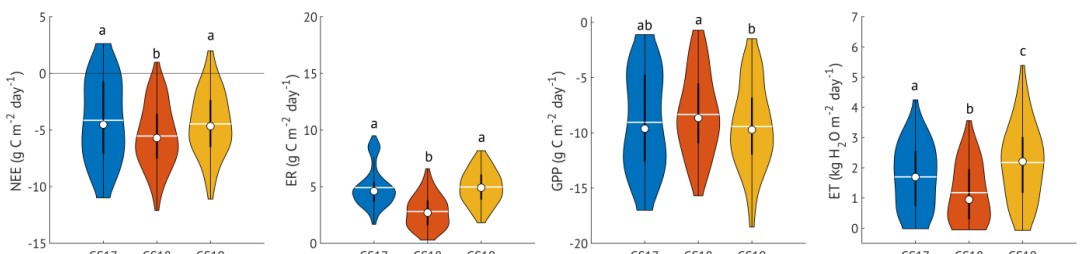

in 2019 and 36% lower than in 2017). Total ET was also the smallest in 2018 (42% lower than 2019 and 23% lower than 2017). Similar to the annual EWUE dynamics, active vegetation season EWUE was the highest in 2018, followed by a 36% decrease in the following year. ET to precipitation ratio was 0.98, 0.76, 0.83, in May-September of the three studied years, respectively.

**Figure 3: Violin plots of carbon and water fluxes over the three studied growing seasons. Markers are median values, and white horizontal lines denote averages. Colours denote growing seasons of different years. Matching letters mark no statistically significant difference between the medians (Wilcoxon signed-rank test p>0.05, Appendix table 1)**

Daily NEE during the GS exhibited the highest net C uptake rate in 2018 (Fig. 3), while no significant difference was observed between daily NEE in 2017 and 2019 (p = 0.2). Daily ER in GS2018 was significantly reduced compared to both other years; daily GPP in GS2018 was similar to the previous GS but smaller than the following year's GS. ET across all three GSs differed significantly, with the highest ET recorded in 2019 and the lowest in 2018, resulting in the highest total EWUE in GS2018 (Table 1).

## 3.3 Seasonal dynamics of carbon and water exchange

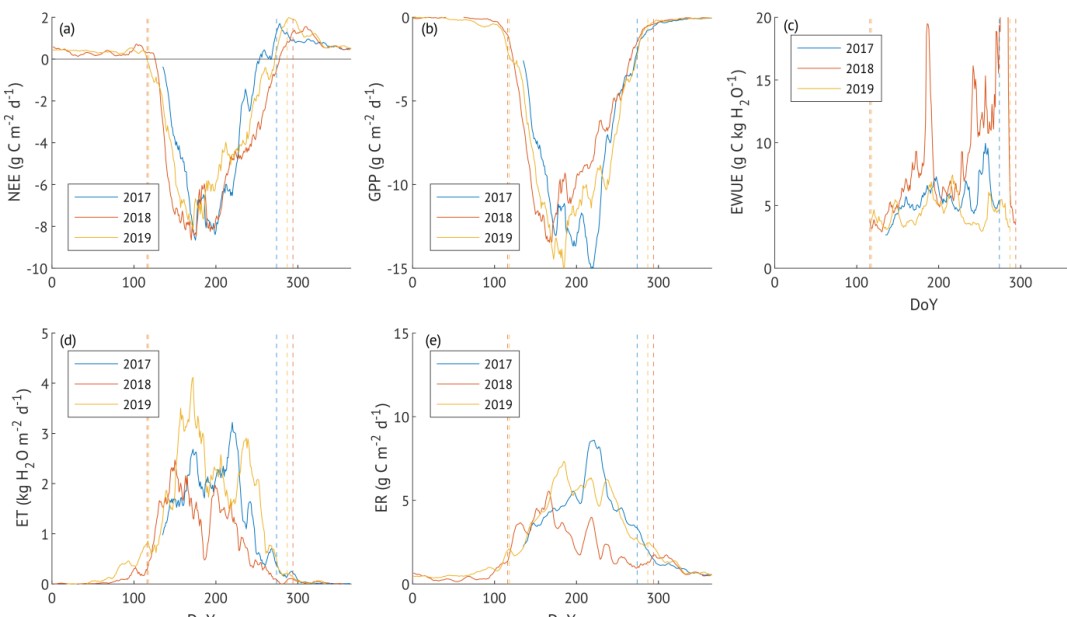

**Figure 4: Seasonal dynamics of net ecosystem exchange (NEE, a), gross primary production (GPP, b), ecosystem water use efficiency (EWUE, c), evapotranspiration (ET, d), ecosystem respiration (ER, e), and represented by 10-days running means. Vertical dashed lines are the borders of growing seasons (GS). EWUE was calculated from 10-days running means of GPP and ET.**

In all three study years, daily NEE was positive (net C release) during the late autumn, winter and early spring and predominantly negative (net C uptake) from the end of April - beginning of May (Fig. 4a). NEE peaked around the second part





of June, followed by a slight decrease in sink strength and a second peak around mid-July, observed in 2017 and 2018, but not

in 2019. Daily NEE reached -8.9, -8.8 and -7.4 g C m$^{-2}$ d$^{-1}$ in 2017-2019, respectively. The ecosystem transitioned to a consistent net C source by mid-September 2017 and by the end of September 2019. In 2018, NEE remained negative for the longest period, reaching positive values only in early October. The autumn months were characterised by net C release reaching 1.3, 1.4 and 1.7 g C m$^{-2}$ d$^{-1}$ in the three study years, respectively.

The seasonal cycle of ER varied across all three years (Fig. 4e), reaching its maximum in the early part of August 2017 (8.3 g

C m$^{-2}$ d$^{-1}$) and the latter part of June 2019 (6.9 g C m$^{-2}$ d$^{-1}$). In 2018, the initial increase of ER that peaked at 4.5 g C m$^{-2}$ d$^{-1}$ in the beginning of summer was followed by a rapid decrease mid-June and significantly ($p<0.0001$) lower values throughout the entire GS.

The GPP seasonal dynamics was more consistent between 2018 and 2019, with the peak occurring mid-June 2018 and in the beginning of July 2019 (Fig. 4b). However, absolute GPP values were smaller at the start of GS2019 and higher throughout

most of the GS compared to the previous year. The GPP dynamics in 2017 differed from the two subsequent years, starting with lower absolute values but becoming more prominent in the second half of GS, peaking later in the first part of August. The highest C uptake was -14.6, -12.8 and -13.5 g C m$^{-2}$ d$^{-1}$ in 2017-2019, respectively.

The ET seasonal cycle exhibited a clear pattern with low values outside the GS and two distinctive peaks during the GS of all the years; however, their height, timing, and the dip between them varied considerably (Fig. 4d). ET reached maximum of 3.1

kg H$_2$O m$^{-2}$ d$^{-1}$ in the beginning of August 2017; the highest ET of 2018 was in the end of May beginning of June (2.4 kg H$_2$O m$^{-2}$ d$^{-1}$), and the 2019 ET peaked the most in the end of June-beginning of July (3.5 kg H$_2$O m$^{-2}$ d$^{-1}$). EWUE reached its highest values in 2018, marked by a notable peak in the beginning of July and substantially higher values in the second part of GS compared to other years (Fig. 4c). EWUE varied the most in 2018 (CV=2.02), followed by 2017 (CV=0.25) and 2019 (CV=0.23).

**3.4 The impact of environmental drivers on carbon and water fluxes**

Across the GSs of all three studied years, water and C fluxes increased with air temperature, although the shape of response curves varied (Fig. 5 a1 – a4). In 2017 and 2019, NEE and GPP displayed similar patterns with no impact (NEE) or slight increase (GPP) with air temperature up to around 10 °C, followed by a sharper rise and a saturation at approximately 22 °C. In 2018, GPP reached the saturation point earlier (19°C) and exhibited no further impact until 27 °C, after which it was slightly

reduced. Generally, GPP, ER and ET had lower values across all air temperature bins in 2018. More negative NEE at lower air temperatures in 2017 was caused by increased GPP compared to other years, while ER was similar to 2019. ET at lower temperatures was higher in 2017 compared to the other two years. The correlation between the C and water exchange and the air temperature was significant in most of the years, except for ER in GS2017, when soil temperature was indicated as the main factor, and ET when the air temperature effect was overshadowed by the leading influence of radiation (Appendix table

250  2).

Atmospheric dryness demonstrated a saturating effect on NEE, caused by GPP saturation at VPD around 7-8 hPa for all the years, although 2018 GPP was lower across most bins. Due to the non-monotonic nature of VPD impact on GPP (the decrease after the plateau at around 14 hPa), $r_s$ was very low or not significant (Appendix table 2). ER increased with VPD in 2018, likely due to its connection with temperature, while no impact of VPD on ER was observed in 2017 or 2019. A similar,

although less sharp, saturating effect of VPD was observed for ET, with 2017 and 2019 exhibiting almost identical curves; the 2018 VPD response curve was lower and flatter. VPD was the second most important environmental driver for ET (after Rg) in all years ($r_s$ ranging from 0.30 to 0.45) and the most important when the Rg was considered as a fixed factor ($r_s$ 0.37-0.54).




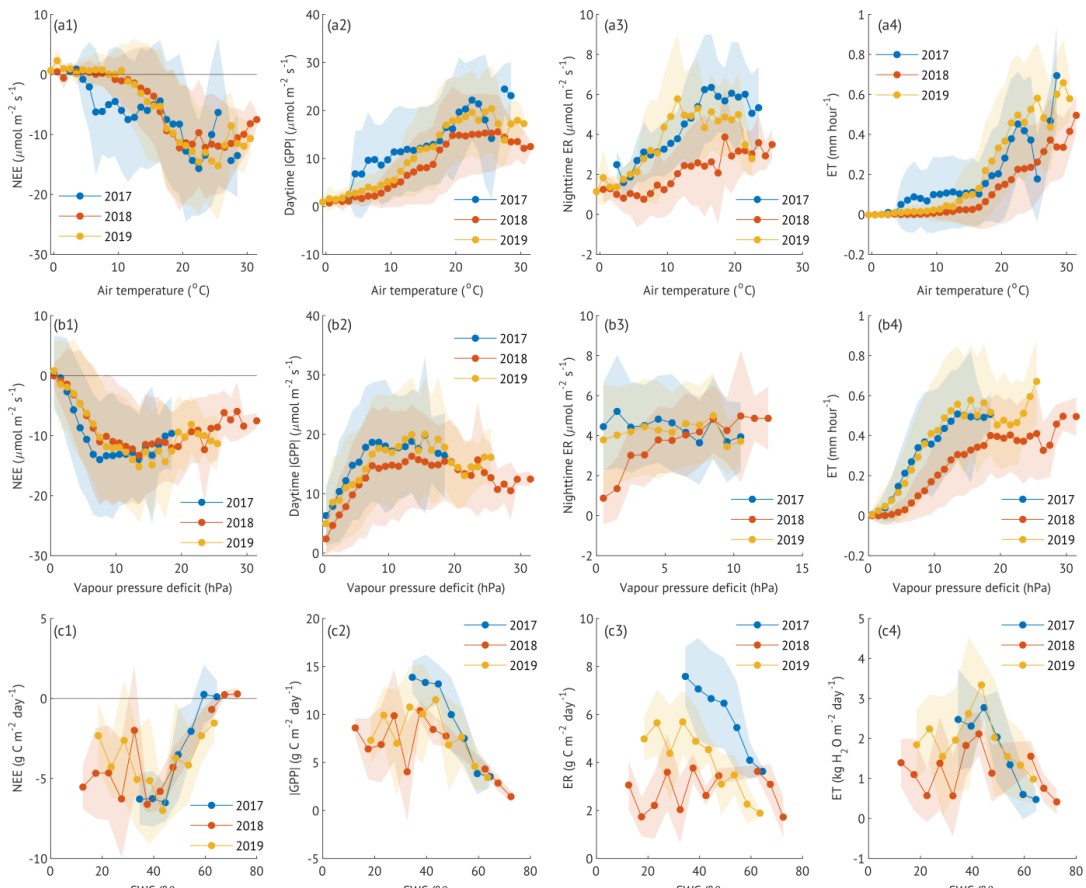

**Figure 5: The influence of air temperature (a1-a4) and vapour pressure deficit (VPD, b1-b4) at a half-hourly scale and soil water content (SWC, c1-c4) at daily scale on carbon and water fluxes in 2017-2019 growing seasons. The absolute values of GPP are used in a2, b2 and c2 for convenience. Markers represent the averages of 1 °C / 1 hPa / 5% bins of Tair, VPD and SWC, respectively. Shaded areas denote the standard deviation.**

Both NEE and ET reached an optimum at around 40% of SWC. Lower SWC resulted in high variability of NEE stemming from the differences in ER, while GPP in 2018 and 2019 was similar under these lower SWC conditions). SWC was identified as the leading driver for GPP in each GS when the impact of Rg was controlled for ($r_s$ = -0.34 … -0.51).

**Table 2. Parameters of light and temperature response curves (estimates and standard errors per growing season)**

| Growing seasons | α | GPP$_{max}$ | ER$_{day}$ | $R^2$ | ER$_{10}$ | $R^2$ |
|---|---|---|---|---|---|---|
| | μmol J$^{-1}$ | μmol m$^{-2}$ s$^{-1}$ | μmol m$^{-2}$ s$^{-1}$ | | μmol m$^{-2}$ s$^{-1}$ | |
| **2017** | 0.12 ± 0.01 | -28.68 ± 0.69 | 3.24 ± 0.61 | 0.44 | 2.08 ± 0.13 | 0.43 |
| **2018** | 0.08 ± 0.01 | -28.65 ± 0.76 | 3.16 ± 0.42 | 0.46 | 2.19 ± 0.07 | 0.11 |
| **2019** | 0.15 ± 0.01 | -26.58 ± 0.50 | 4.38 ± 0.56 | 0.47 | 3.01 ± 0.09 | 0.24 |

The response of GPP to Rg over all three GSs was consistent, with 2017 and 2018 exhibiting similar light response curve parameters. The GS of 2019, however, demonstrated a slightly lower maximum GPP (GPP$_{max}$) and higher daytime ER (ER$_{day}$). ER$_{10}$ had a similar pattern with the highest value occurring in 2019.




**3.5 Photosynthetic capacity and water use efficiency**

The annual canopy photosynthetic capacity (GPP$_{sat}$) was -36.94, -34.77, -36.44 µmol m$^{-2}$ s$^{-1}$ in 2017-2019, respectively. GPP$_{sat}$ exhibited a sharp increase with the highest absolute values at by the end of May – beginning of June (DoY 173-190) followed by a midsummer reduction in the second part of July – beginning of August, observed in all the years (DoY 192-225, Fig. 6a),

however it much shorter in 2017. GPP$_{sat}$ was steadily decreasing from the end of August. While the spring 2017 had lower GPP$_{sat}$ and not so pronounced depression, no significant difference was found in GPP$_{sat}$ between the GSs of the three studied years (Appendix table 1).

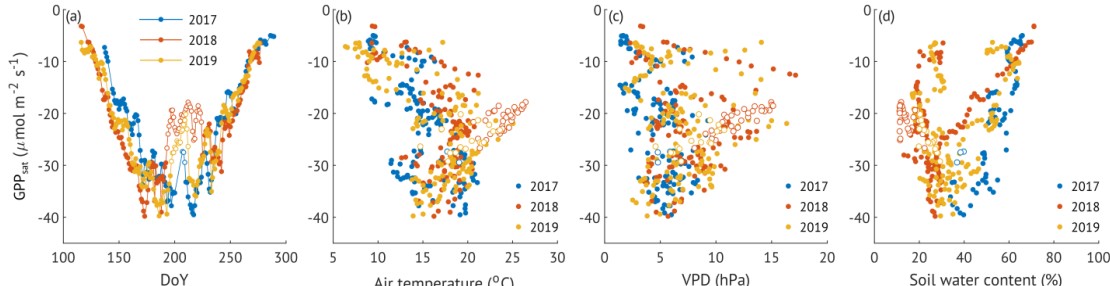

**Figure 6: Seasonal cycle (a) of and environmental drivers' influence (b-d) on GPP$_{sat}$ during the three measurement years. White markers denote the midsummer GPP$_{sat}$ reduction in all sub-figures**

Midsummer reduction of GPP$_{sat}$ in 2018 was accompanied by increased air temperature (period average and standard deviation: 23.9±1.5 °C), high VPD (12.1±1.2 hPa) and very low SWC (14.4±3.4%) (Fig. 6b-d). A shorter reduction period in 2019 had moderate to high air temperatures (20.0±2.4°C), moderate VPD (8.0±1.8 hPa) and decreased SWC (24.3±3.3%). Only three 5-days periods in 2017 featured significant light response curve fits with reduced GPP$_{sat}$. They were characterised by moderate air temperature (18.4±0.7°C), low VPD (5.1±0.6 hPa) and moderate SWC (38.3±1.5%).

Daily values of EWUE varied greatly with single outstanding values corresponding to high VPD and moderate air temperature. In 2018, EWUE was heightened at lower VPD values compared to other years. Partial correlation analysis (Appendix table 2) also identified VPD as the main driver ($\rho$ =-0.36… - 0.56), closely followed by SWC in 2017 ($r_s$ = -0.32) and 2018 ($r_s$=-0.23) with no correlation in 2019. Air temperature demonstrated a positive partial correlation in all three years ($r_s$ = 0.16 – 0.33).

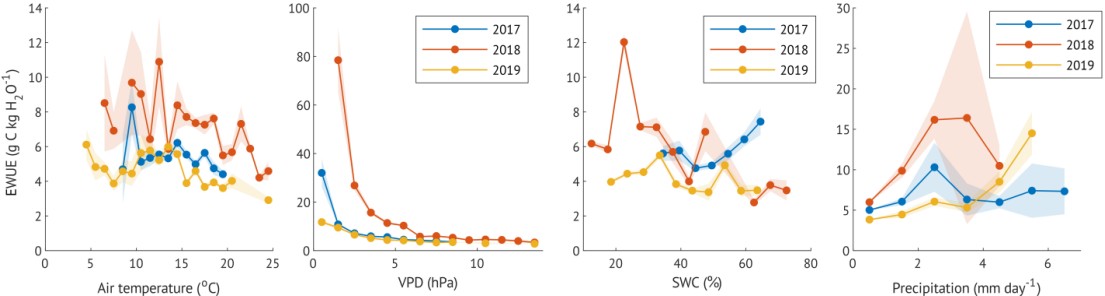

**Figure 7: The impact of air temperature (a), vapour pressure deficit (b), soil water content (c) and precipitation on EWUE. EWUE**
**was calculated as the ratio of bin-averaged daily GPP to bin-averaged daily ET, based on the averages of 1-degree Tair bins, 1hPa VPD bins, 5% SWC bins, 1mm precipitation bins with the minimum of three days per bin. The bin size was chosen based on the data availability. Shaded area denotes standard error.**

**4. Discussion**

While our study covered only three years, the contrasting environmental conditions provided a unique opportunity to assess C
and water exchange under distinct moisture regimes. The study years ranged from a wet year (2017) to a drought year (2018) and an intermediate year (2019) in terms of SWC and air temperatures, additionally allowing for an assessment of potential



short-term legacy effects. Despite pronounced differences in meteorological conditions, the site remained a strong net C sink throughout the study period, with annual NEE ranging from -496 to -663 g C m$^{-2}$ y$^{-1}$. Notably, the warmest and driest year, 2018, exhibited the highest net C uptake, a pattern driven by enhanced GPP in spring and suppressed ER in late summer and
autumn months.

Interannual differences in water fluxes were more pronounced. ET was the lowest in 2018, leading to a markedly higher EWUE, which increased by 40% relative to the average of the other two years. The reduction in ET, coinciding with elevated VPD, suggests that stomatal regulation constrained water loss under drier conditions. Seasonal patterns further revealed a midsummer decline in canopy photosynthetic capacity (GPP$_{sat}$) in all years, with the most pronounced reduction occurring in
2018 when SWC was at its lowest.

Taken together, these results indicate that the grey alder forest maintained high C uptake efficiency under hot and dry conditions, likely due to the combined effects of reduced respiratory losses and optimized water use. However, the enhanced midsummer depression in GPP$_{sat}$ suggests that photosynthetic activity was nonetheless constrained during peak drought periods, highlighting physiological trade-offs under moisture-limited conditions. These findings highlight the capacity of
riparian grey alder forests to function as persistent C sinks even under variable climate conditions, yet also underscore the importance of evaluating their long-term resilience under increasingly frequent climate extremes.

### 4.1 Carbon balance

Alder is a widely distributed tree species across hemiboreal and temperate zones, commonly found in riparian buffers, yet data on its forest C exchange remain surprisingly scarce. In a chronosequence of alder forest stands, studied by Uri et al. (2017),
the oldest forest ("Kalliste") was of similar age as the current site at the time of the research (40 years old). The total annual net ecosystem production (NEP) of Kalliste stand was -77 g C m$^{-2}$ y$^{-1}$ denoting the site as a weak net C source ecosystem (NEP = -NEE, Chapin et al., 2011), while the average of -586 g C m$^{-2}$ y$^{-1}$ makes our site a strong sink of C. The differences in C balance between the sites likely stem from differences in soil fertility, as the Kalliste stand was established on former grassland, whereas the present site is located on nutrient-rich former agricultural land. These findings highlight the role of soil fertility in
determining forest C sequestration potential.

Compared to previously reported values for broadleaved forests in boreal and hemiboreal zones, the NEE at our site exceeds most estimates but aligns with fluxes observed in more southern broadleaved and coniferous forests (Table 3). While GPP at our site was comparable to that of boreal and hemiboreal forests, ER was notably lower. In contrast, forests with a similar NEE range (for example, in southern Sweden, Denmark, and Germany) exhibited higher GPP but also greater ER, likely driven by
their warmer and sunnier climate. Riparian forests, such as our study site, receive substantial inputs of leaf litter and organic material, yet decomposition rates can be constrained by wet soil conditions. In waterlogged or anoxic layers, organic matter breaks down more slowly, potentially contributing to lower ER. Additionally, alder forests are known for their rapid growth and high nitrogen cycling (Aosaar et al., 2012; Rytter and Rytter, 2016; Uri et al., 2017), which may enhance GPP without necessarily accelerating decomposition if soil moisture remains high. On the other hand, reduced soil water availability during
the drought year appeared to suppress heterotrophic respiration while GPP remained mostly unaffected. Rapid fluctuations in SWC, that are characteristic for riparian ecosystems, could potentially dampen decomposition rates, leading to lower annual ER.

**Table 3. Comparative table of net ecosystem exchange (NEE), gross primary production (GPP) and ecosystem respiration (ER) from various broadleaf forests and forests with values close to this study. All numbers are in g C m$^{-2}$ y$^{-1}$**

| Site description | NEE | GPP | ER | Reference |
|---|---|---|---|---|
| Mature alder forest in Estonia | -586 ± 85 | -1330 ± 82 | 743 ± 166 | This study |





| Mixed broadleaf forest in boreal Canada | -80 … -290 | - | - | (Black et al., 2000) |
|---|---|---|---|---|
| Oak forest in boreal Canada | -206 ± 92 | -1343 ± 85 | 1171 ± 139 | (Beamesderfer et al., 2020) |
| Alder/Ash mixed forest in Germany | -193 | -1595 | 1401 | (Kutsch et al., 2005) |
| Beech forest in Denmark | -313.6 … -353.8 | -1977.4 … -2302.4 | 1663.8 … 1948.6 | (Lindroth et al., 2020) |
| Oak-dominated forest in Germany | -559 | -1794 | 1235 | (Kutsch et al., 2005) |
| Spruce forest in Germany | -663 ± 78 or -535 ± 72 * | -1680 ± 103 or -1755 ± 249 * | 1020 ± 106 or 1219 ± 232 * | (Ney et al., 2019) |
| Beech forest in France | –386 ± 171 | -1347±192 | 1011 ± 138 | (Granier et al., 2008) |
| Spruce forest in Southern Sweden | -192.9… -582.3 | 1851.6… - 1869.0 | 1286.8 …1658.7 | (Lindroth et al., 2020) |
| Pine forest in Estonia | -214 ± 113 | -1264 ± 49 | 1050 ± 118 | (Rogozin et al., in print) |

*depending on the gas-analyser heating correction

**4.2 Water exchange and water use efficiency**

Distinct seasonal patterns in ET in our study were shaped by the interplay of key environmental factors, including Rg, VPD, air temperature and precipitation (Brümmer et al., 2012; Jassal et al., 2009; Massmann et al., 2019). The close alignment between the seasonal cycles of ET and GPP further supports the long-established coupling of plant water and C exchange

through stomatal regulation (Jarvis, 1986). The mid-season decline in ET, which coincided with a similar reduction in GPP, was likely a response to lower VPD and diminished solar radiation – both identified as primary regulators of ET (Jassal et al., 2009). This decline was particularly pronounced in 2018, when prolonged drought conditions and limited precipitation further constrained ET. The bell-shaped response of ET to SWC resulted in reduced total water fluxes in growing seasons with both high (2017) and low (2018) SWC, whereas total ET peaked in 2019 under intermediate soil moisture conditions. These findings

underscore the dual influence of atmospheric and soil moisture controls on ET dynamics and highlight the sensitivity of alder forest water fluxes to interannual variability in hydroclimatic conditions.

The evapotranspiration-to-precipitation ratio (ET/P) provides further insight into the site's water balance and its response to changing hydroclimatic conditions. During the May-September of 2017 (the "wet" year), ET nearly equalled precipitation (ET/P = 0.98), suggesting that the most of precipitation was used for transpiration and soil and wet surface evaporation, with

minimal excess contributing to runoff or deep percolation. In contrast, the same months of 2018 (the drought year) exhibited the lowest ET/P ratio (0.76), indicating a precipitation surplus and constrained water loss, likely due to stomatal closure in response to soil moisture depletion and high VPD. The ET/P ratio increased again in 2019 (ET/P = 0.83, in May–September), suggesting a partial recovery in transpiration as soil moisture availability improved. This interannual variability highlights the forest's capacity to adjust water use under different climatic conditions, with a clear suppression of ET during drought and a

subsequent increase as conditions became more favourable.

The EWUE in our study was notably higher than reported for various forest ecosystems globally. The average EWUE of 5.3 ± 1.2 g C kg $H_2O^{-1}$ exceeded values observed in mixed temperate forests (1.9-4.1 g C kg $H_2O^{-1}$; Jin et al., 2023), deciduous





forests in the USA (2.3-2.7 g C kg $H_2O^{-1}$; Xie et al., 2016) and Central China (2.6 ± 0.7 g C kg $H_2O^{-1}$; Niu and Liu, 2021), as well as the global range for forested ecosystems (0.8-3.6 g C kg $H_2O^{-1}$; Zhou et al., 2014), suggesting that high EWUE may

be a characteristic feature of nitrogen-fixing riparian forests. The ability of the alder forest to maintain elevated EWUE, suggests an efficient water conservation strategy that supports sustained C assimilation under varying moisture conditions. This trait may enhance the ecosystem's resilience to future climatic extremes, reinforcing the potential role of grey alder forests in maintaining regional C sinks under shifting hydroclimatic regimes.

### 4.3 Drought impact and the absence of legacy effect

Despite the 2018 heatwave, the grey alder forest remained a strong C sink, exhibiting the highest net C uptake of the study period. In spring, increased GPP drove a higher net uptake, while in late summer and autumn, suppressed ER was the primary contributor to enhanced NEE. Warmer spring temperatures have previously been shown to stimulate net C uptake by extending the growing season (Keenan et al., 2014; Wolf et al., 2013) as was also observed in our study, offsetting the influence of the forthcoming summer drought on the annual C balance (Angert et al., 2005; Kljun et al., 2006; Smith et al., 2020; Wolf et al.,

2016). A similar pattern was reported for a floodplain mixed broadleaf forest in the Czech Republic, where an anomalously warm spring in 2018 led to an increase in both GPP and ET, counteracting the negative effects of the summer drought (Kowalska et al., 2020). In boreal and hemiboreal regions, moderate warming in spring typically coincides with sufficient soil moisture availability from snowmelt, ensuring adequate water supply for early-season C assimilation. However, enhanced spring productivity and transpiration can also accelerate soil water depletion, increasing susceptibility to summer drought

stress (Bastos et al., 2020).

The impact of 2018 drought on various Nordic forests was analysed by Lindroth et al. (2020). In a beech forest, the only broadleaved forest included in their analysis, both GPP and ER decreased by approximately 300 g C $m^{-2}$ $y^{-1}$, with GPP experiencing a slightly stronger reduction, leading to a minor decrease in annual NEE. In contrast, the forest in our study exhibited a much smaller annual change in C fluxes, with GPP and ER declining by only 52.5 and 116.2 g C $m^{-2}$ $y^{-1}$,

respectively. The stronger suppression of ER compared to GPP was likely the key factor maintaining high net C uptake in 2018.

Water fluxes were more strongly affected by the drought. ET was significantly reduced (for 35%) in 2018, leading to the highest EWUE of the three study years (40% higher than the average of the other two years). EWUE is often used as an indicator of a forest's ability to optimize C assimilation under changing water availability (Huang et al., 2015; Keenan et al.,

2013; Yang et al., 2016). Maintaining or increasing EWUE during unfavourable or extreme conditions provides the ecosystem with a sufficient reserve of carbohydrates, which may later facilitate recovery. Conversely, less flexible ecosystems may experience C deficiency that could also be reflected in subsequent years (Frank et al., 2015; Kannenberg et al., 2020). An increase in EWUE during drought, as observed in our study, has been previously reported for a boreal aspen stand in Canada (Krishnan et al., 2006) and a mixed deciduous forest in Switzerland (Wolf et al., 2013). However, responses appear to be

species- and site-dependent; for example, no change in EWUE was observed in a Finnish forest under low rainfall conditions (Ge et al., 2014), while a decline in EWUE was reported for a pine forest in Finland under severe drought stress (Gao et al., 2017). These contrasting patterns highlight the importance of species-specific drought adaptation strategies and site hydrology in determining forest water use responses.

The midseason reduction in canopy photosynthetic capacity ($GPP_{sat}$) under high temperatures and low soil moisture suggests

physiological constraints on photosynthesis under limiting conditions. A similar, though less pronounced, reduction was observed in 2019, pointing to a potential legacy effect. However, total GPP in 2019 was the highest of the study period, and the concurrent increase in ET led to a lower EWUE. Combined with higher ER, these findings suggest that the ecosystem was in a recovery phase rather than experiencing prolonged drought-induced C limitations. Moreover, the difference between



GPP$_{sat}$ of all three studied years was not significant, denoting alder forest under study as a functionally stable ecosystem (Chen et al., 2023).

In contrast, strong legacy effects on the C cycle have been observed following the 2018 drought in other European forests. For example, in a mixed deciduous forest in central Germany, NEP declined by 150 g C m$^{-2}$ y$^{-1}$ in 2019, with reductions in both GPP (-281 g C m$^{-2}$ y$^{-1}$) and ER (-132 g C m$^{-2}$ y$^{-1}$) compared to the previous year (Pohl et al., 2023). European beech forests have exhibited particularly high sensitivity to drought, with observed tree mortality linked to hydraulic failure (Rukh et al., 2023; Schuldt et al., 2020). More broadly, drought-induced tree mortality can have long-lasting consequences, with post-drought effects often persisting for months or years (Brodribb et al., 2020; Schwalm et al., 2017). A global synthesis showed that drought legacy effects are widespread in forests, typically lasting three to four years (Anderegg et al., 2015), with post-drought temperature and precipitation conditions strongly influencing recovery time (Schwalm et al., 2017). Drought-related growth decline and canopy dieback have also been documented in various riparian trees (Kibler et al., 2021; Singer et al., 2013; Stella et al., 2013; Valor et al., 2020). In our study, we found no visual or statistical evidence of increased tree mortality in the year following the 2018 drought. However, given that drought-induced mortality can manifest with a delay, it remains possible that long-term effects could emerge beyond the period of our study. Future monitoring would be critical to assessing whether the observed recovery is sustained or whether cumulative drought stress could compromise forest resilience over time.

### 4.4 Sustainability and forest management considerations

The benefits of grey alder in forestry are well recognized across the Nordic and Baltic regions, where it is valued for its rapid early growth, high productivity, nitrogen-fixing capacity, broad ecological range, frost resilience, and relatively low susceptibility to pests. The recommended harvesting age for grey alder stands is typically between 20 and 25 years (Uri et al., 2014), although significant increases in woody biomass continue beyond this age. For instance, Aosaar et al. (2012) suggested that the optimal age for harvesting could be closer to 40 years. Our findings demonstrate that even at this advanced age, the riparian grey alder forest remained a strong C sink, showing no clear evidence of a drought legacy effect following the 2018 heatwave. However, as drought-induced mortality can occur with a delay, a single post-drought year may be insufficient to assess potential long-term resilience. While the forest is considered ready for harvesting, the trade-off between maximizing short-term timber revenue and sustaining long-term C sequestration remains uncertain. These considerations highlight the need for longer-term research, particularly on the C balance and drought resilience of riparian alder forests beyond 40 years of age, to guide both sustainable management strategies and climate mitigation efforts.

### 5. Conclusions

The mature riparian grey alder forest remained a strong and consistent net C sink across three years with contrasting soil moisture conditions. The highest net C uptake in 2018, despite the heatwave and drought, was driven primarily by suppressed ER in response to moisture limitation, with only a minor impact on GPP. Similarly, ET was significantly reduced, leading to a 40% increase in EWUE. While photosynthetic capacity (GPP$_{sat}$) declined during the peak drought stress, there was no significant difference between the years, suggesting functional stability.

The absence of a clear drought legacy effect in 2019, combined with the forest's ability to sustain high EWUE and net C uptake under extreme conditions, suggests that riparian grey alder forests are highly adaptable to short-term hydroclimatic variability. Unlike other European broadleaved forests where long-lasting drought impacts have been observed, this riparian alder stand maintained its productivity and resilience. However, as drought-induced tree mortality can manifest with a delay, longer-term monitoring would be essential to assess whether these forests remain resilient under increasing drought frequency and severity.





Balancing long-term C sequestration with sustainable forest management remains a key challenge. While the forest in this study has reached a typical harvestable age, the potential trade-offs between timber production and long-term C uptake warrant

440 further investigation. Future research should examine how stand age, site conditions, and climate extremes affect the stability of alder forests over time, as well as explore the effects of alternative management strategies, such as extended rotations or mixed-species planting, on maintaining resilience under a changing climate.

**Appendix**

**Appendix table 1. Bonferroni-adjusted p-values of Wilcoxon signed-rank test comparison between the growing seasons of different**
445 **years.**

| Growing seasons | NEE | ER | GPP | ET | EWUE | GPP capacity |
|---|---|---|---|---|---|---|
| 2017 vs 2018 | 2.42E-07 | 2.60E-16 | 0.220742 | 2.44E-05 | 0.000242 | 1.3101 |
| 2018 vs 2019 | 5.60E-07 | 4.67E-19 | 0.001133 | 2.25E-11 | 1.35E-07 | 0.9263 |
| 2017 vs 2019 | 0.200246 | 1.243908 | 0.261011 | 0.000288 | 0.01033 | 0.1189 |

**Appendix table 2. Spearman's coefficients ($r_s$) from partial correlation analysis between half-hourly (ER, GPP, ET) and daily (EWUE) values and environmental drivers. Only statistically significant ($p<0.05$) results are presented**

| | Nighttime ER | | | Daytime GPP | | | | | | | ET | | | | | | | EWUE** | | | |
|---|---|---|---|---|---|---|---|---|---|---|---|---|---|---|---|---|---|---|---|---|---|
| Year/driver | Ta | Ts | SWC | Rg | Ta | SWC | VPD | Ta* | SWC* | VPD* | Rg | Ta | SWC | VPD | Ta* | SWC* | VPD* | Rg | Ta | SWC | VPD |
| 2017 | | 0.38 | | 0.72 | 0.29 | -0.37 | -0.09 | 0.46 | -0.51 | 0.18 | 0.63 | | 0.12 | 0.45 | 0.15 | | 0.46 | | 0.29 | -0.32 | -0.56 |
| 2018 | 0.24 | 0.08 | 0.18 | 0.70 | 0.15 | -0.20 | -0.15 | 0.22 | -0.34 | | 0.33 | | 0.10 | 0.30 | 0.21 | | 0.37 | 0.28 | 0.16 | -0.23 | -0.36 |
| 2019 | 0.21 | 0.08 | | 0.69 | 0.46 | -0.24 | -0.37 | 0.38 | -0.40 | -0.03 | 0.62 | 0.05 | 0.12 | 0.41 | 0.42 | 0.14 | 0.54 | | 0.33 | | -0.40 |

*The correlation while controlling for global radiation (Rg)
450 ** Based on daily values

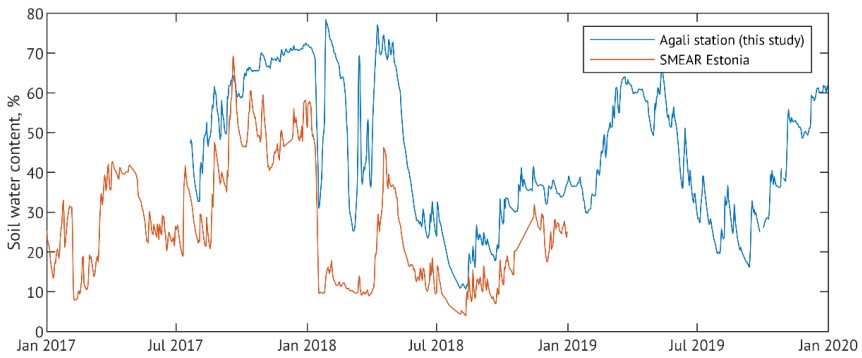

**Appendix figure 1. Soil water content at the study site (blue) and SMEAR Estonia station (red)**

**Data availability**

The data are available upon request from the authors.



**Author contribution**

Article conceptualization was done by AK with the help of KS and ÜM; data calculation and processing were performed by AK with valuable input from SR and DK; KS and ÜM acquired the funding and managed the Agali site; AK prepared the figures and wrote the article with editorial contributions from all authors.

**Competing interests**

The authors declare that they have no conflict of interest.

**Acknowledgements**

This study was supported by the Estonian Research Council (IUT2-16, PRG-352 and PRG2032), the European Union Horizon programme under grant agreement No 101079192 (MLTOM23003R), and the European Research Council (ERC) under grant agreement No 101096403 (MLTOM23415R).

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
