# Peer review of "Mature riparian alder forest acts as a strong and consistent carbon sink"

_EGUsphere, 2025_

## Author Response (AR1)

Dear Editor and Reviewers,

On behalf of all co-authors, I would like to thank you for your time and effort and for the valuable and comprehensive comments provided. We believe that the manuscript has improved substantially following the implementation of most of your suggestions. While we tried to shorten the text whenever possible, however, new parameters and analyses still expanded the manuscript length.

As the text was extensively rewritten, it is not feasible to provide a marked copy; however, we indicate the relevant line numbers to help locate the changes.

The main revisions, made in accordance with the reviewers' feedback, are summarised below:

- **Introduction**: shortened and rephrased for improved clarity; objectives updated accordingly.
- Methods: expanded to include additional site information, parameters, and analyses:
  - Added information on soil properties of the study site (Section 2.1).
  - Provided more detail on the calculation of canopy physiological response parameters and included an additional parameter, canopy conductance (*Gc*) (Section 2.4).
  - Fully revised the analysis of soil moisture impacts, incorporating soil saturation ratio and reference canopy conductance analyses (Section 2.5), with particular emphasis on distinguishing the effects of SWC from other environmental drivers, especially VPD.
  - Added a more comprehensive analysis of drought recovery (Section 2.6).

**- Results:**

- completely rewritten to merge overlapping content and remove unnecessary repetition;
- o energy balance closure results added (details in Appendix A);
- o precipitation data was updated with data from a nearby station
- o all figure colours updated with a more inclusive palette;
- o The text per section was shortened, and unnecessary details were omitted.
- **Discussion and Conclusions:** largely rewritten to integrate the results of the new analyses. Land-use policy and climate mitigation implications were removed to keep the study more focused.

Below, we provide our detailed, point-by-point responses following the major revision, with Reviewers' original comments marked in *italic*.

**Anonymous Referee #1**

Dear Authors and Editor.

In general, I find the manuscript interesting. The methodology is sound, and I see merit in the study. However, I am concerned that the manuscript is, at times, overly lengthy and difficult to follow, which makes it hard to read overall. In several sections, critical information is either difficult to locate or entirely absent.

**Response:** Thank you for your positive assessment of our methodology and the overall merit of the study. When revising the manuscript, we paid extra attention to improving conciseness and clarity throughout.

I believe the authors could present more concisely what I see as the key result of this study: why and how evapotranspiration (ET) and gross primary production (GPP) decouple under anomalously dry conditions.

**Response:** Thank you for this helpful observation. While we did not originally describe our findings using the term "decoupling," we agree that the different responses of ET and GPP during the drought year are one of the important results. In particular, GPP remained relatively stable while ET dropped, which led to a strong increase in EWUE. We highlighted this observation in lines 616-624

Moreover, the Discussion section often repeats similar sentence structures (e.g., "These findings/results highlight that...") and reiterates basic, well-established principles of ecosystem functioning. This repetition detracts from the overall readability and does not add new insights.

**Response:** We appreciate this feedback and have revised the Discussion to remove redundant phrasing and avoid unnecessary repetition of well-established concepts.

The authors conduct numerous comparisons with other broadleaved forest ecosystems, which, I assume, are mostly not riparian systems. They attribute differences in net ecosystem exchange (NEE) or other variables to various factors such as soil nutrient availability or vaguely defined climatic variability. These comparisons sometimes feel overly detailed and only loosely connected to their own results. I recommend refining this section to focus on comparisons that directly support their findings.

**Response:** Thank you for this comment. Our aim with the comparisons was to provide context for the magnitude of carbon fluxes observed at our site, particularly given the

scarcity of published NEE values for riparian alder forests. In the revised manuscript, we have shortened this section (lines 504-509) and strengthened the discussion by expanding the comparison with other alder forests (lines 482–503).

Although I understand that specific data on other GHG fluxes (e.g.,  $CH_4$  and  $N_2O$ ) are not available for the site under study, I believe the authors should at least qualitatively discuss these fluxes. Making educated assumptions about their potential roles in riparian ecosystems would strengthen the manuscript's conclusion that riparian alder forests could contribute meaningfully to climate mitigation through carbon sequestration.

**Response:** Thank you for this suggestion. While other GHGs were not the focus of this manuscript, the measurements of  $CH_4$  and  $N_2O$  were conducted at the same site in 2018 and 2019 (and published in Mander et al., 2021,2022), and we agree that including a brief discussion of these gases strengthens the interpretation of riparian alder forests in the climate mitigation context. The site was found to be a net methane sink in 2018-2019 and a source of  $N_2O$ . However, the  $N_2O$  emissions were relatively small and offset only a small share of the net  $CO_2$  uptake (when expressed in  $CO_2$ -equivalents). These findings are now summarised in the revised manuscript (lines 520–527).

Despite these issues, I find the paper interesting and within the scope of Biogeosciences. It should be considered for publication after thorough revisions. I have included specific comments below, but I would like to emphasize that the authors should carefully revise the arguments, logic, and structure, particularly in the Results and Discussion sections, to improve readability and clarity in the next version.

**Response:** We thank the reviewer for this encouraging evaluation and hope that the revised manuscript reflects improved clarity, coherence, and scientific focus.

**Specific comments:**

**Introduction:**

The conclusions of Paragraphs 1 (l.26) and 2 (l.36) could be more clearly distinguished. As written, both highlight the need for monitoring with respect to drought response and carbon sink capacity, but without clearly separating their specific focuses (i.e., short-term drought effects vs. long-term C-sink function).

**Response**: We agree that the conclusions of these two paragraphs were not clearly distinguished. We have rewritten both paragraphs (lines 25-34)

Line 50: You state that "few studies" exist, but provide no citations. Does this imply that no studies have been published yet? Please clarify or provide supporting references.

**Response:** Thank you for this observation. We have clarified this statement in lines 53-54

Line 54: It would be helpful to briefly explain what is meant by "the traditional C budgeting method" to provide context for readers unfamiliar with the term.

**Response:** Thank you for this suggestion. We have added more information about the carbon budgeting methods and differences between them in the Introduction (lines 55-59) and further elaborated in the Discussion (lines 484-485). We have also recalculated values from one method to the other for improved comparability, as detailed in Appendix D.

Line 53: This paragraph is somewhat unclear. You suggest that climatic factors outweigh stand age in importance, yet both old and young stands are described as carbon sources. Does this imply differing climatic conditions between the sites? Please clarify. If the implication is that the sites differ in climate, that should be explicitly stated. Furthermore, without detailed knowledge of the study by Uri et al. (2019), one could infer that its findings, based on "nutrient-rich former agricultural land" (l.319), may not be broadly applicable to typical riparian alder forests, which are unlikely to share these conditions. While this may not be the case for your study, the question arises whether your results are representative or overly site-specific. Consider expanding this paragraph or the relevant discussion section to clearly position your site within the broader context of existing research, particularly when findings from other studies appear to diverge.

**Response:** Thank you for this detailed and helpful comment. In the revised manuscript, we have shortened this section in the Introduction (lines 55-59) and expanded the relevant part of the Discussion (lines 482–503) to better position our site within the broader context of existing research. We have also added more information on the sites' soil properties (Table 2) to support the comparison.

Lines 60f: If you cite your previous research as a foundation here, please briefly summarize its key findings. This paragraph is currently difficult to follow. It doesn't explain how forests responded to a heatwave, nor why that is relevant to the current manuscript—aside from the mention that water fluxes were not considered previously. Consider including a paragraph that outlines previous findings and highlights the open questions your study aims to address. Then, consider reformulating your objectives for increased precision. In particular, Objective 2 ("different soil moisture regimes and WUE") may already be included in Objective 1 ("quantify... water exchange... under varying hydroclimatic conditions

**Response:** Thank you for this suggestion. We rewrote this section and updated the objectives. See lines 59-64

**Methods:**

Figure 1: The overview figure is very informative. As there is free horizontal space, consider including an additional ground-level photo, perhaps of the instrumentation setup or the canopy. This would help everyone unfamiliar with the ecosystem visualize the site.

**Response:** Thank you for this thoughtful suggestion. Unfortunately, we could not find a photo with sufficient resolution, and the measurement setup has since been relocated to another site.

Line 79: Minor detail—consider removing the term "total" since you're only showing the 80% footprint. Out of curiosity: do the remaining 15% correspond to the river and the water bodies?

**Response:** We agree that the term "total" was misleading and have removed it (line 94).

Lines 79f: It would be useful to include information on variation in stand height, stem diameter, etc. From the image in Figure 1, the stand appears relatively uniform. Quantifying this would strengthen your argument.

**Response:** Thank you for this suggestion. Unfortunately, we do not have more detailed information on the stand's structural parameters beyond what is already reported.

Lines 105f: This statement is surprising, as the manuscript has not yet introduced the discrepancy between soil and EC fluxes. Please clarify or provide context earlier.

**Response:** We agree that the reference to a discrepancy between soil and ecosystem-scale fluxes appeared abruptly without prior context. This paragraph has been rewritten for clarity (lines 121-130).

Lines 122f: Consider briefly explaining your rationale for using the nighttime flux partitioning method (e.g. over the daytime method).

**Response:** We chose the nighttime-based flux partitioning method because it relies on directly measured ecosystem respiration, is widely used and appropriate for our relatively flat site with low nighttime advection after filtering. The rationale has now been added to the Methods section (lines 148–151).

Lines 129f: This paragraph feels a bit too short. A rationale for the analyses would be helpful. If it's too lengthy for the Methods section, it could be placed earlier near the objectives

**Response:** This section has been fully rewritten to incorporate the additional parameter (canopy conductance, Gc) as suggested by Reviewer 2. We have also improved its structure and clarity, See section 2.4

Around Equation 1: It appears there's an issue: the equation references  $ER_{day}$  in the text, but this term doesn't appear in the equation itself. Additionally, please explain how canopy light

use efficiency was calculated and which variables were used. Lastly, please clarify what modifications were made to the model and cite its original source, not just studies that have used it.

**Response:** The ERday was indeed incorrectly mentioned in the explanatory text around Equation 1, it will be removed. The canopy light use efficiency ( $\alpha$ ) was obtained as a fit parameter of the rectangular hyperbolic light-response curve (Equation 1), which relates gross primary production (GPP) to incoming global radiation (Rg). This model was applied using a 5-day moving window on daytime half-hourly values with Rg > 15 W m-2. The curve-fitting was performed using non-linear least squares, and only results with statistically significant fit parameters (p < 0.05) and R2 > 0.5 were retained.

The model applied is a form of the original rectangular hyperbola by Michaelis and Menten (1913), without the ERday term, since it was already accounted for during the flux partitioning. We chose this form over the non-rectangular variant (e.g., as used by Musavi et al., 2017; Chen et al., 2023) because it resulted in a greater number of valid fits across all years in our dataset. The corresponding section has been rewritten for clarity (lines 172–183).

Lines 143f: How exactly are start and end of the growing season defined - using a relative or absolute GPP threshold? Consider also explaining why growing season length is relevant to your analysis. Later (e.g., l.176), you note that growing season lengths do not differ significantly between years, but it's unclear whether or how statistical tests were applied here. Please clarify. If the differences are statistically insignificant and not central to your main conclusions, you might consider shortening this section.

**Response:** The start and end of the growing season were determined using a curve-fitting approach, not an absolute or relative GPP threshold. Specifically, we applied the double-logistic fitting method to daily GPP values, following Gonsamo et al. (2013). The inflexion points of the fitted curve were used to define the start and end of the GS. These details have been added (lines 156–158). As growing season length was not a primary focus but rather a means to define periods for calculating canopy physiological response parameters, this section has been shortened to a single sentence.

**Results:**

Figure 2 (and all other figures): It looks like standard color palettes were used. Please ensure the color schemes are accessible to readers with color vision deficiencies. If needed, use colorblind-friendly palettes or add alternative line styles. Also, clarify whether panel d) represents net radiation or incoming radiation. Minor suggestion: you might consider omitting the year legend repetition across all panels as readers can refer back easily once it's introduced.

**Response:** Following your suggestion, we updated the colour schemes of all figures to a palette that passed accessibility testing for colour vision deficiencies. We thank the reviewer for this comment, as the revised figures are now clearer and more informative. We also removed unnecessary legend repetitions and clarified axis labels.

Line 144: Consider using a different abbreviation for the correlation coefficient than " $r_s$ ," as it could easily be mistaken for stomatal conductance commonly used term in flux studies.

**Response:** We used "rs" to denote Spearman's correlation coefficient, following common statistical notation. However, this analysis has been removed from the manuscript,

Lines 251f.: This paragraph is difficult to follow. It relies heavily on the brief mention of partial correlation analysis back in line 144, which readers are unlikely to remember without very close reading. Please reintroduce the purpose, method, and results of this analysis in a self-contained way here. A visual representation such as a scatterplot of the residuals (just one of many possibilities) might help as well. Currently, readers will likely jump to Figure 5 and find your interpretation hard to align with what's shown. In this context, consider moving key results from the appendix table 2 into the main text, possibly in a visually more appealing way.

**Response:** This analysis has been removed from the revised manuscript.

Line 264: The statement that "SWC was the leading driver..." is a bit surprising given that Fig. 5c2 doesn't clearly support this. Is this conclusion mainly due to radiation?

**Response:** The conclusion was based on the partial correlation analysis controlling for radiation, where SWC showed the strongest correlation with GPP across all years. As this analysis has been replaced, the revised results are now presented in Section 3.4

Table 2: As mentioned earlier, please clarify the analysis conducted here. Also, since the table isn't referenced in the main text, its purpose and contribution are unclear—please address this.

**Response:** Thank you for this observation. Table 2 has been removed from the manuscript.

Line 269: You state that GPPmax was "slightly lower" in 2019, but Table 2 shows a difference of  $\sim$ 0.07, which is negligible. Please clarify this wording or interpretation.

**Response:** Table 2 has been removed from the manuscript.

**Discussion**

Line 315: Introducing NEP here is slightly confusing. Consider converting their values to NEE for consistency and improved readability.

**Response:** We retained NEP here to remain consistent with the original paper. However, we agree this may cause confusion, as NEP is not used elsewhere in the manuscript. We have replaced NEP with NEE.

Line 330: You suggest that rapid SWC fluctuations might reduce annual ER, shouldn't this be testable based on your dataset, or is something missing?

**Response:** Thank you for this valuable suggestion. While rapid SWC fluctuations may influence decomposition and thus ER, our dataset does not allow direct testing of this hypothesis. Soil and air temperatures, strong drivers of ER, likely mask the subtler effects of SWC variability, and the absence of heterotrophic respiration measurements prevents us from isolating decomposition rates from total ecosystem respiration. We acknowledge this limitation, and the corresponding point has been removed from the manuscript.

Table 3: In the entry for the Swedish spruce forest (Lindroth et al., 2020), the GPP value is missing a minus sign. Overall, the formatting in Table 3 is inconsistent. Some entries include  $\pm$  values, others list single values, and some show a range ("..."). There are also inconsistencies in spacing and in the use of decimal places versus integers. A general reformatting would help improve clarity. As currently formatted, it's unclear whether you're showing interannual variability or uncertainty, or whether "..." denotes a range. Please clarify.

**Response:** The table has been reformatted for consistency and clarity. The updated table now appears as Table 3 with consistent notation for uncertainty and decimal places.

Line 337: Small note: The current sentence structure implies a direct connection between your results and the cited studies. Consider rephrasing for example: "Similar to other studies (e.g., Xy et al., Yz et al.), we observed that seasonal ET patterns were shaped by..." or omit the references if they are not directly aligning with your results.

**Response:** We agree with this comment and revised the wording in Discussion.

Line 338: This statement is confusing. Your growing-season data (e.g., Fig. 3) shows ET is reduced during the drought year while GPP remains largely stable.

**Response:** Our reference to the close alignment referred to the overall seasonal patterns of GPP and ET, rather than a direct indication of their quantitative coupling. However, this paragraph has been removed from the revised manuscript.

Line 340: The sentence starting with "The mid-season decline in ET..." feels awkward, shouldn't your analysis directly address and explain this pattern?

**Response:** We agree with this comment; however, this paragraph has been removed from the revised manuscript.

Line 350: A reported precipitation surplus in this riparian setting is surprising, given typically high evaporation. This is quite interesting, consider elaborating further.

Response: We appreciate the reviewer's interest in this observation. While riparian systems are typically associated with high evapotranspiration, the apparent precipitation surplus during the 2018 drought likely reflects reduced plant water use due to stomatal closure under high VPD and soil moisture depletion. However, in the absence of runoff or drainage measurements, we cannot fully quantify the water balance, and our interpretation remains speculative. Furthermore, the ET fluxes presented were not corrected for lack of energy balance closure – that is, we did not adjust latent and sensible heat fluxes to match available energy, as measurements of net radiation and ground heat flux were not available. Although this may lead to an underestimation of absolute ET, we consider the interannual comparisons to remain valid, given that the methodological approach was applied consistently across years. We have clarified these limitations in the Methods (lines 132–135) and elaborated on the low ET values in the Discussion (Section 4.2).

Line 373: Just a curiosity in this context: how significant is soil water depletion between spring and summer in riparian systems? A brief discussion could be insightful.

**Response:** Soil water depletion between spring and summer in riparian systems can vary considerably depending on groundwater connectivity, precipitation patterns, and vegetation water use. In systems with strong hydrological connectivity to groundwater, depletion may be minor; however, under drought conditions or in systems with limited lateral or vertical recharge, significant drawdown can occur. At our study site, we observed a clear seasonal decline in topsoil moisture during summer, suggesting that even in this riparian setting, soil water depletion was substantial under dry conditions. This point is now briefly discussed in the revised manuscript (lines 642–647).

Line 377: You note that both GPP and ER decreased by 300 g C—is this the same amount for each, and does that mean NEE?

**Response:** We thank the reviewer for spotting this potential error. This sentence has been removed from the revised manuscript.

Lines 394f: The discussion around the lack of a legacy effect and its occurrence in other ecosystems feels somewhat lengthy and secondary to your core findings—but I may be missing the relevance. You demonstrate that physiological stress was present but relatively moderate, and that the ecosystem adapted and recovered quickly. This might be the key takeaway here. For readers less familiar with Nordic ecosystems, it might help to contextualize the severity of the drought in climatological terms. For example., was it a 10-year drought, 50-year event, etc.?

**Response:** We agree that the recovery is a particularly interesting finding, as previous studies have reported stronger drought impacts in the year following the event. We have revised the discussion to emphasise this resilience (See Section 4.5). We also thank you for highlighting the missing information on drought severity. We added the sentence in the introduction (line 39)

**RC2: 'Revealing the means and variability of C and water fluxes of under-studied hemiboreal alder forest: potential but deeper analysis necessary', Samuli Launiainen**

Dear Editor and Authors,

here my review on "Mature riparian alder forest acts as a strong and consistent carbon sink" by Krasnova et al., https://doi.org/10.5194/egusphere-2025-1280

The study quantifies ecosystem-atmosphere carbon (C) exchange and evapotranspiration of a fertile riparian alder forest established on former agricultural land in Estonia by statistically analyzing nearly three years of eddy-covariance (EC) measurements. The data from the mature (ca. 40 yrs of age) hemiboreal forest covers the European 2018 heatwave, giving an opportunity to assess the response of alder forest to extreme hydroclimatic variability. The studied ecosystem shows reduced gross-primary productivity (GPP) and ecosystem respiration (ER), and improved ecosystem water-use efficiency (EWUE) during the most intensive drought period. On annual timescale, however, the net ecosystem productivity (NEP) was strongest during the drought year, and no significant carry-over effects on C exchange were observed in the first post-drought year. This suggests the alder ecosystem C balance is resilient to droughts, and that compensatory mechanisms (e.g. earlier growing season start etc.) during the year can have stronger impact on annual C sink than relatively short-term hydrological extremes.

Standard EC methodologies are used throughout, and the study design, data curation and applied flux gap-filling and partitioning methods seems sound. The only exception is that energy balance closure should be shown for each of the three growing seasons to increase the confidence on the low evapotranspiration (ET) values reported.

**Response:** Thank you for this valuable suggestion. We agree that assessing energy balance closure is important to support the interpretation of the observed low ET. Unfortunately, a rigorous evaluation is not possible due to limitations in our measurement setup - we lack direct measurements of net radiation and ground heat flux. Nevertheless, we performed an approximate closure assessment for each of the three growing seasons using the available data. The results are presented in Appendix A.

The study aims to reveal the inter-annual variability between hydrometeorologically contrasting years, particularly the effect of soil moisture content (SWC) and atmospheric dryness (vapor pressure deficit VPD) on ecosystem C flux components and water use

characteristics. The study provides rather unique dataset from European hemiboreal alder ecosystem, and the scope fits that of Biogeosciences.

The main problems with the current manuscript (MS) are: 1) The statistical analysis applied are not well-suited to address and separate the impacts of SWC and VPD on other variability, whether due to seasonal cycle or due to correlations of these soil and atmosphere dryness-metrics with other environmental variables. 2) Because of this, the MS is too descriptive, and Discussion contains too many vague arguments that are not backed up with in-depth analysis or literature. Combined with some overly detailed (!) and repetitive parts in the Results section, this makes the MS a bit frustrating to read and it is hard to gasp the key points.

Overall, there is potential and the study can be a useful addition to the literature. However, additional analysis is needed to better reveal the short-term response of the ecosystem to progressing 2018 drought, and subsequent recovery. The discussion can also be easily improved by better usage of literature to interpret the observed changes via physiological and biogeochemical knowledge. Some concrete suggestions of potential analysis are given below in the Detailed comments. I do not expect the authors to do all of them but provide them rather as ideas how to strengthen the analysis.

**Response:** We would like to thank the reviewer for their detailed and constructive feedback. We agree that, in its initial form, the manuscript was overly descriptive in some parts, and it benefited from a more focused, more causal analysis. A summary of the major changes is provided at the beginning of this document.

**Detailed comments:**

L16: unclear to which time period reported ecosystem WUE represents.

**Response:** The reported values in lines 14-17 are annual values, however, they were removed from the abstract

L19: what is 'in response to H20 limitation'? Do you mean response to VPD, soil moisture availability or the combined effect?

**Response:** Here, we meant soil moisture. However, the sentence was removed in the revised manuscript

L23-24: Natural climate solutions were not focus of the study and not addressed at all

**Response**: We agree, this part has been omitted from the revised manuscript.

L31-33: Sentence is vague and has no information; rephrase

**Response:** We agree, that the original sentence is too vague. The corresponding paragraph has been rewritten (lines 25-34)

L55: C sinks, while a young...

Response: Thank you

L64-65: Here and especially in results and discussion, the authors should pay more attention on the relevant timescales of the responses. Throughout the paper, it is often very unclear whether annual, seasonal or short-term variability (e.g. how fluxes respond to progressing soil water limitations, and how they recover after rainfall) is discussed. This is a major issue, and should be better addressed in the revised version

**Response:** Thank you for your observation. In the revised manuscript, we have paid extra attention to the timescales under study

L67: avoid buzzwords 'nature-based solution for climate mitigation' OR significantly deepen the discussion on the potential (i.e. impact, scalability) of using alder forests to improve land C sink on former agric. lands / and or to optimize riparian zone management.

**Response:** We appreciate your suggestion. In earlier drafts, we considered discussing the potential of alder forests as a nature-based solution for climate mitigation. However, as the manuscript evolved, the focus shifted more strongly toward characterizing the impact of environmental factors on water and carbon fluxes. To maintain a clear and consistent narrative, we decided to omit broader implications related to land-use policy and climate mitigation.

L71: The site is on former agricultural land, so land-use history may have strong effect on soil C storage and thereby ER?

L78-84: Information on ecosystem leaf-area index (LAI), and site land-use and forest management history are missing. Are the above- and belowground C stocks quantified elsewhere?

**Response:** We agree with both comments above and acknowledge that land-use history can strongly influence soil carbon storage and ecosystem respiration. We added more information on the site's history and soil properties (see Section 2.1). Unfortunately, LAI data are not available for this site.

L99: Be consistent with terms NEP (used in introduction) and net ecosystem exchange (NEE)

**Response:** We agree with this comment, and NEE is now used throughout the revised manuscript.

L114-115: Bad sentence; I assume these percentages represent the data coverage?

**Response:** Thank you for pointing this out. The sentence was unclear, and we agree that clarification is needed. The percentages refer to the proportion of half-hourly NEE values that remained after quality control. The sentence has been clarified (lines 129-130)

L132: Calculation of EWUE needs more details. Was it estimated on 30min basis or from accumulated fluxes? What are the relevant timescales reported? Does it represent dry-canopy conditions or all conditions, and what are the impact of this choice on (physiological) interpretation of the results? If comparing the effect of soil drought on EWUE, shouldn't you cluster the data into similar radiation and VPD conditions?

**Response:** Thank you for your valuable suggestions. Our initial approach to calculate EWUE at different time scales indeed led to a more descriptive paper and complicated the analysis of environmental drivers' impact. Thus, we have implemented the following steps in the revised manuscript:

- 1. EWUE was recalculated using either full sums (for yearly and seasonal values in Table 1) or aggregated values of GPP and ET utilising half-hours under sufficient light conditions and restricted calculations to dry, active-canopy days within the growing seasons to obtain canopy EWUE (lines 161-169). The "sufficient light" threshold was determined using breakpoint analysis to identify the flattening point of the light response curve (lines 158-161). This way, we reduced the impact of light on canopy EWUE variability in the following analysis.
- 2. To assess drought effects, we used VPD-normalised canopy EWUE values, bin-averaged by soil saturation ratio (line 212)

L139 (eq. 1): Definition of ER\_day is not relevant here

**Response:** The ERday was indeed incorrectly mentioned here and was removed.

L143-144: The growing season was defined using GPP; thus it corresponds to carbon uptake period.

**Response:** In this context, we used the term "growing season" as defined in the original methodological paper by Gonsamo et al. (2013), which is based on GPP dynamics. Our intent was to capture the period of active photosynthesis, corresponding to canopy activity. The term "carbon uptake period" can refer either to the period of nonzero GPP or negative NEE (i.e., net carbon uptake), making it more ambiguous in this context.

L149-151: Sentences are unclear, and I wonder if using constant activation energy (E0) is consistent with the nighttime flux-partitioning method used?

**Response:** We thank the reviewer for this observation. We agree that the original wording was unclear. To clarify: in our application of the Lloyd and Taylor equation, the activation energy parameter  $(E_0)$  was estimated separately for each year, while the reference

temperature (Tref) was set to 15 °C, and T0 was kept constant at -46.02 °C. This approach is consistent with the standard implementation of the nighttime flux-partitioning method.

L160 Figure 2 panel (f): add cumulative precipitation on right y-axis; current barplot is not informative alone.

**Response:** We appreciate this comment. The figure was updated to highlight the progressive drought as a result of dry spells, especially in July 2018.

L176-178: The study focuses on inter-annual variability. What controlled the IAV of growing season length; is early onset of growing season related to high air temperature?

**Response:** We agree that higher air temperature is likely the primary driver of earlier growing season onset. However, our dataset is limited in this regard: measurements in 2017 began relatively late, so we effectively have reliable estimates of growing season start only for two years. This sample size is too small to draw statistically robust conclusions on the controls of inter-annual variability in growing season length. We omitted the part about the growing season length differences, since it was not the focus of the paper.

L217-218: Example of vague text: what is meant by late autumn, winter and early spring?

**Response:** Thank you for this example. In the revised manuscript, we tried to improve the wording and be more precise.

L220 & L222-223: What time periods the daily average NEE refers to? Is this information necessary for the study goals?

L224-227: Example of unnecessary repetition of figures. Please interpret the figures causally using e.g. Fig. 2 instead of repeating their content. Same concerns to large extent whole section 3.3; that there is seasonal cycle in C uptake and ER is not particularly new. Consider merging Sect 3.3 and 3.4 to better link the changes in ecosystem fluxes to their drivers, to reduce repetition and to improve the clarity.

L233-239: Link ET variability to weather variability and plant phenological stage (LAI development) rather than repeat the figure in text.

**Response:** We agree that the original Results were overly descriptive. The entire Results section was rewritten to reduce repetition, focus on causal interpretation, and merge sections 3.3–3.4 for clarity.

L251-251: At which timescale and period? Core growing season or throughout? To demonstrate the significance of stomatal control on GPP (and NEE) further, compute surface conductance (Gs) from measured ET, cluster it to conditions with ample light and show the dependency of Gs to VPD. You may see different shape of Gs-VPD curve or dropping reference conductance (Gs\_ref) when soil is dry? Oren et al. (1999; https://doi.org/10.1046/j.1365-3040.1999.00513.x) model Gs/Gs ref = -m \* ln(VPD), where

Gs\_ref is reference conductance at VPD=1kPa and  $m\sim0.6$  provides theoretical grounds to compare the observed dependency.

**Response:** We thank the reviewer for this valuable and constructive suggestion! The suggested analysis was added to the manuscript. See lines 184-192; 213-225 in Methods and Section 3.4 of the Results, as well as Appendix A for more details.

Similarly, consider showing how e.g. GPPmax varies during progressing drought.

**Response:** Thank you for this suggestion! The analysis of soil moisture impact on GPPsat was added, see lines in 202-205 Methods and Section 3.4 of the Results.

L253: rs is not defined

**Response:**  $r_s$  here is a partial correlation coefficient. However, partial correlation analysis was omitted from the manuscript

L259 Figure 5: Nice figure but interpreting the responses of NEE, components and ET (y-axis) to single environmental factors (x-axis) is complicated because you consider the whole growing season, meaning that e.g. high soil moisture conditions represent spring and autumn (Fig. 2). Same concerns temperature and VPD responses.

As you have evaluated GPPmax in a moving window, consider sub-setting the data so that you include only 'the stable summertime when the canopy is fully developed'. This will enable better insights on the role of VPD and soil moisture as controls of ecosystem behavior? I also suggest you explicitly show the response of NEE, GPP, ER and EWUE to VPD and soil moisture over 2018 (and maybe for other years as well) while selecting only conditions with ample light and temperature constrained to a narrow range (i.e. avoiding extremes?). Also, omit rainy periods.

**Response:** Thank you for this helpful observation! We agree that including the entire growing season in Figure 5 complicated the interpretation of flux responses to single environmental drivers. We significantly updated our soil moisture response analysis. Please see Section 2.5 in the Methods and Section 3.4 in the Results. All the values used in the analysis were filtered for the active season dry-canopy periods with sufficient light conditions.

L272: What is meant by 'annual photosynthetic capacity'?

**Response:** As mentioned in the methods section, annual photosynthetic capacity is 95th percentile of each year's GPPsat. However, these values seem meaningless for the overall discussion, so they have been be omitted.

L277 Figure 6: Also here seasonal variability and short-term variability are mixed in panel b-d. What is the main message of this figure? Can it be improved e.g. by showing different years

with different symbols and adding day of year as a color scale?
L289 Figure 7: Again seasonal variability overshadows responses to drought? In last panel, high EWUE occurs during rainy days in 2018. Are you sure this is not an artifact of underestimated ET measurement when the canopy is wet?

**Response:** Thank you for these constructive comments. We agree that in both original Figure 6 and Figure 7, seasonal variability and short-term variability were confounded, making it difficult to isolate the effects of drought or other environmental drivers. They were removed.

L301-311: This part would benefit significantly from separating seasonal cycle from more short-term drought impacts. Strengthen the arguments by use of literature.

**Response:** This part was omitted from the manuscript

L315-320: There is nearly order-of-magnitude difference in the net C sink of these two alder forests. Different land-use history is one plausible reason, but this is presented as a hypothesis as no references are given? At L323 it is noted that ER of the studied ecosystem is lower than comparable boreal and hemiboreal forests. Can this be due to the land-use change and depleted soil C storage – what does the literature tell us?

**Response:** Thank you for pointing this out. We expanded the comparison (lines 482-503), adding details on soil properties (Table 3) and recalculating the biometrical carbon budget parameters to more comparable values (see Appendix D). We could not find more studies on alder forest carbon exchange, calculated with eddy-covariance method.

L329-330: Two issues: 1) heterotrophic respiration was not quantified and therefore the argument is not backed up with the data. 2) GPP shows significant reduction in 2018 during the dry period (Fig. 4 & Fig 2) compared to other years. On annual / growing season scale GPP was not reduced, likely because of larger uptake in early warm spring season?

**Response:** We agree that the interpretation regarding suppressed heterotrophic respiration during the drought year is speculative. We rephrased this part, see lines. We also acknowledge that GPP did decline during the peak drought period in 2018. It was indeed compensated by enhanced uptake in early spring, resulting in little change in total growing season GPP. This is discussed in lines 579-592.

L330-331: Rapid fluctuations in SWC... this is pure speculation the effect was not addressed.

**Response:** The sentence regarding rapid SWC fluctuations dampening decomposition was removed since it was not tested in the current study.

Table 3: For the Danish beech forest (Soroe), cite original reference Pilegard & Ibrom (2020, https://doi.org/10.1080/16000889.2020.1822063) rather than Lindroth et al. (2020) drought synthesis

**Response:** Thank you for this comment; it was fixed.

L340-346: The data (Fig. 5c4) indicates ET has bell-shaped but scattered response to soil water content (SWC). At the wet end (high SWC) this does not mean excess soil water content or limited oxygen availability would be restrict transpiration, as the conditions with high SWC cluster to early/late growing season days when evaporative demand is low (i.e. low available energy and VPD).

Response of stomatal conductance and thereby transpiration rate (proportional to LAI x gs x VPD) to soil moisture is typically highly non-linear and it would be interesting to see how this manifests itself in the data. If you see clear threshold-type response, that could be used to cluster data to 'no drought' vs. 'water-limited' regimes, to explore how Pmax, Gs, EWUE etc. differ when soil water content is limiting?

In practice: subset data for ample light, no rain, fully developed canopy LAI etc. and show Gs, GPP, ET, ... vs. SWC, or preferably 'soil saturation ratio', i.e. SWC/porosity where porosity ~ upper percentiles of observed SWC.

This is an example of how to move from 'descriptive interpretation of the flux timeseries' into more physiologically relevant impact-analysis. See also earlier suggestion on additional analyses towards this direction.

**Response:** Thank you for this comment and your suggestions. We agree, that the seasonal variability overshadowed the impact of SWC. We have followed your suggestions. See sections 2.5 in the Methods and 3.4 in the Results

L347-352: Can/should you comment on the role of lateral water flows? You study a riparian forests so I assume those can be important for soil moisture dynamics especially in early growing season, leading to delayed depletion of SWC and thus mitigating for late summer drought stress. It is interesting that you still see such a strong drought response in ET.

**Response:** Although our site has a gentle slope of about 1%, lateral water flows can still be relevant, specifically for the riparian ecosystem. However, since we lack the runoff or drainage measurements at our site, we can only briefly acknowledge their potential contribution to soil moisture dynamics in the revised manuscript (lines 642-646)

L353-355: Argument on increasing transpiration may be true but remains fully speculative as ET partitioning was not done.

Response: Thank you for this important point. We acknowledge that without ET

partitioning, statements regarding changes in transpiration remain speculative. While the separation of transpiration and evaporation components was out of the scope of the current study, we base our interpretation on the general understanding of ecosystem water flux responses to drought and stomatal regulation.

L356-363: The EWUE in your study is very high, as shown by comparison with other forests. This is either due to surprisingly tight stomatal control of alder, or due to underestimated ET. Did you check the energy balance closure and evaluate whether the reported (low!) ET values are plausible compared to other forests in similar climate conditions? Can plant trait databases or publications on leaf-level water use efficiency provide support to your interpretation that water use of alder is extremely conservative, i.e. leaf-level IWUE = A/gs is high?

**Response:** Thank you for the valuable comment and suggestions! The EWUE was indeed high, due to the low ET. Here're the updates that we did in response:

- The low ET is discussed in Section 4.2 of the Discussion. The high EWUE was logically discussed there as well
- The energy balance closure is provided in Appendix A. It is indeed at a lower end (70%, 71% and 80% in 2017,2018 and 2019, respectively). While it could impact our ET estimates (which we acknowledge in lines 542-543), it was also estimated with the lack of available energy measurements at the site.
- We added a leaf-level reference, although only one that we managed to find, that actually found on the contrary a higher stomatal conductance for alder leaf with higher transpiration (lines 550-551)

L379: Here and throughout the MS: consider how many significant digits to report taking into account typical uncertainties

**Response:** We agree, and it was fixed

The current analysis is not well suited to detect the legacy effects of the 2018 drought L395: Why this would be a legacy effect and not just a typical response to environmental variability (e.g. VPD and soil moisture) over 2019 growing season?

**Response:** The drought recovery part was updated please see sections 2.6 in Methods; 3.5 in the Results and 4.5 in the Discussion

L397-398: What is meant by recovery phase? Was the ER higher due to environmental conditions, of because of there was excess of undecomposed young litter from the dry 2018 year? Argument is just handwaving.

**Response:** Thank you for your valuable observations. Due to the absence of litterfall measurements, we can only speculate on the potential contribution of excess undecomposed litter from the 2018 drought to the elevated ER in 2019.

L415-417: sentence needs backing from the literature reference.

**Response:** We agree, however, that section was removed from the manuscript.

*L433:* "extreme conditions" --> during limited water availability (or soil drought)?

**Response:** Thank you for this suggestion, the Conclusions were re-written.

---

## Author Response (AR2)

**Authors' response to Anonymous referee #1**

On behalf of the authors, I would like to thank the Reviewer 1 for their time and effort to improve our manuscript. Please find my response below with the Reviewer's original comments marked in *grey italic*. The line numbers refer to the clean version (without the track changes).

While the MS Word automatic track changes document marked all figures as new, we updated only 2 figures:

- Figure 9 was remade for the altered indices (see the explanation in the corresponding section)
- Figure 8 has now corrected the y-axis labels, which were previously mislabelled as "modelled"

Following the first round of review, the authors have made a tremendous effort to revise the manuscript and to address my comments as well as those of the second reviewer, which I highly appreciate. In my view, the manuscript has improved substantially and can now be considered for publication after the authors address a few remaining minor comments, many of which are only suggestions aimed at improving readability and accessibility.

**Response:** We sincerely thank you for your thoughtful comments, which greatly helped improve the manuscript. The study has already improved considerably compared to the original version.

I found one issue regarding the new analysis around Figure 9, where it appears that the authors may, perhaps inadvertently, have changed the definition of certain metrics and applied a different resampling technique than in an earlier part of the manuscript, resulting in diverging results for the same dataset. This is somewhat confusing and should be clarified.

**Response:** Thank you for identifying this inconsistency arising from the new analysis. Your comment helped us carefully reconsider the methodology and what the analysis results actually tell us about our forest. We provide a detailed response and clarification under the specific comment further below.

Despite this, given the overall high quality of the revised manuscript, I recommend to the editor that it be accepted after the authors address the comments at their own discretion, without another round of review. I have high confidence in the authors' ability to do so and look forward to seeing the final version of the manuscript, which I consider a valuable contribution to the field. Their study skillfully explores the carbon dynamics of a relatively underexplored ecosystem, offering insights that will benefit future assessments.

Below are some specific comments, listed roughly in order of appearance. All line numbers are referring to the manuscript version WITHOUT track changes.

Regarding the whole ms: for carbon fluxes, you use both µmol C m-2 time-1 and g C m-2 time-1 throughout the manuscript. Unless you have a good reason to do so, why not make it consistent?

**Response:** We intentionally used both  $\mu$ mol C m-2 s-1 and g C m-2 (period-1) to match the conventions of flux data presentation at different temporal scales. Parameters, calculated from half-hourly carbon fluxes (GPPsat - canopy photosynthetic capacity; and ERref – respiration at reference temperature) usually retain the standard flux unit of  $\mu$ mol C m-2 s-1, while daily and cumulative (seasonal or annual) sums are expressed in g C m-2 (period-1).

l.63: minor and optional comment: you could change the wording here from "did not address" to "was beyond the scope" as currently it sounds a bit negative towards your own work, which I find unnecessary.

**Response:** We appreciate the suggestion. Although we do not consider "did not address" to be negative, we agree that "was beyond the scope" provides a smoother phrasing. The text has been revised accordingly (line 63-64)

l.66: "We utilise three years of EC flux measurements, representing a "wet" year (2017), a "drought" year (2018), and a "recovery" year (2019)."; My suggestion is that you add something in the tone of: "...three years of EC flux measurements with contrasting environmental conditions: 2017 being an anomalous wet (add numbers here), 2018 an anomalous dry (add numbers here) and 2019 being a recovery year (add numbers here)." Regarding what I mean here by numbers see my comment regarding l.260

**Response:** We agree that highlighting the contrasting environmental conditions adds clarity, however as this sentence is part of the introduction, we prefer to keep it descriptive. See modified sentence in lines 66-67

l.104 & l.107: The tower was 21 m high, but PAR was measured at 25 m — might this be a typo? Also, in Appendix A you state that measured incoming shortwave radiation was used, whereas in the instrumentation paragraph only a quantum sensor is mentioned. Perhaps that information is just missing?

**Response:** Thank you for noticing this inconsistency! This text remained from an earlier version of the manuscript and was not updated. We have now corrected it (lines 105-107).

l.132: I agree with that choice. Later on, I added a comment suggesting that you likely overestimate Rn, which further complicates any potential attempts at closure.

**Response:** While the approach is simplified, it allowed checking if the difference in evapotranspiration between the years would be biased by the EBC variability.

l.241: "filtered to maximise the share of T" This probably means that you filtered based on radiation > X or GPP > 0 values? Maybe you can say what you did here.

**Response:** The filtering criteria have now been specified in the Methods text (lines 250-252)

l.260: This comment is regarding the whole paragraph and also relates to my comment in l.66: I think you could briefly present an argument for the wet/dry/recovery classification. Currently, you describe that the drought and recovery are different because of longer dry periods in the drought year but that none of the years is significantly drier in comparison to the long-term average precipitation. I agree with your classification looking at the values in Table 1 and Fig.2e but I think some kind of quantitative measure would strengthen the manuscript and make this more intuitively. A simple way to do so would be to use ETCCDI indices, which might better show differences between the years. Also, you could mention the above average temperature during the drought period. Taken together, while I don't think it is strictly necessary to do so, I think that currently it's not super easy to figure out by briefly looking at Fig2e together with you saying that the precipitation input is not different and that SWC is mainly governed by precipitation how this classification is justified. In other words, the classification made only sense for me once I understood all the results and it seems not very intuitive yet.

**Response:** We thank the reviewer for this suggestion. We agree that the difference in precipitation is not immediately apparent from the figure. While the total precipitation in 2018 remained within one standard deviation of the multiyear average, it was still lower than in the other years. Its temporal distribution, extended consecutive dry days combined with above-average temperatures ("heatwave"), likely drove the drought conditions. We edited this part in the manuscript (lines 274-288). We hope that this explanation clarifies the wet/dry/recovery classification without adding additional metrics to our already lengthy manuscript.

l.284: Maybe add "cumulative" to the bracket such as: (cumulative annual NEE < 0) and add to the sentence whether the number is the average of all years or the number for 2017 as the brackets have the numbers for the other two years.

**Response**: We agree. The revised sentence in 1.290**

Regarding the values shown Table 1: For SWC you note that measurements started only in July, but Fig. 3 shows that all fluxes also start in May. I think for consistency you could do the same for the carbon fluxes in the table simply referring to what you explain starting in l.183

**Response**: Thank you for this comment. We improved the clarity of the Table 1 caption text and marked the annual 2017 values in italic.

l.300f. minor comment just for consistency: You present very detailed p-values as smaller than, greater than or equals here but in l.306 you say (not significant) which would, for example, be also the case for l.303 (...while GPP declined only marginally). This statement in particular, you could also say something like "GPP did not significantly decline" to match the wording in l.305 where you say that there was "no differences between the daily

values" based on your statistical test. I recommend you check that throughout the manuscript to have more consistency.

**Response**: Thank you for this comment. We edited the paragraph (lines 307-312) and checked for the consistency

Fig. 8: The caption could be slightly improved by more clearly stating that solid lines represent measured data, whereas dashed lines show modelled data. This is also is true for the text, for example in line.450: where you say "GPP in 2018 was reduced from July onwards" add precisely what you mean for easier understanding. Here is another example of what I mean: In the drought year 2018, measured GPP (Fig.3c, Fig.8a) was reduced from July onwards [...] When applying the model parameters from 2018 [...], this GPP suppression persisted in the modelled GPP for both the other years (Fig.8b). I think if you guide the reader a bit more along your results together with the figures it's much easier to follow. In the next sentence ("The difference between the observed GPP..."), I am also not sure if you are referring to all results from this analysis or just to the what we see in Fig 8a and c. This goes on for a bit in this paragraph, the results are very interesting but with small changes it could be much easier to read and understand.

**Response:** We agree that both the Figure 8 caption and the corresponding text in the Results section needed clarification. We have revised the figure caption and rewritten the related paragraph to explicitly state that solid lines represent observed fluxes and dashed lines represent cross-year modelled fluxes. In addition, we corrected the y-axis labels in Figure 8, which were previously mislabelled as "modelled." We would like to note that, as daily aggregated GPP and ER are, strictly speaking, not directly measured, we use the terms "observed" and "cross-year modelled" to distinguish between fluxes derived from gap-filled EC data and those modelled using parameter sets from other years.

Fig.9 and corresponding texts parts (from l. 239 and l.460): I tried a while to understand what the benefit to the overall story of the manuscript of this new analysis is but to be honest couldn't quite figure it out. Some points which I thought some time about

- I looked at the Lloret et al. (2011) paper to understand it and found that what you define as Recovery is equivalent to their Resilience. I don't know if this happened by accident or if you had a certain motivation to do so, if that is the case, please address in the methods section why you made new definitions here. Your resilience term I find quite hard to interpret with the text you provide and I don't know exactly what I should take from it.

**Response:** We very much appreciate this comment! The previous modification of the metrics was a result of our overthinking, which did not improve clarity as intended. After careful consideration, we have reverted to the more conventional definitions: Rt = dry/ref; Rc=rec/dry; Rs = rec/ref. (lines 241-248). Corresponding updates have been made to Figure 9 and throughout the text. Importantly, we note that this revision does not change the results or the conclusions of our study.

- Also, I don't understand the reasoning behind your analysis that you calculating these terms based on daily values. As for example the resistance term you define is just comparing the drought to the reference term that value is equivalent to what you show in and after Table 1, where you, for example, find that the GPP term is insignificantly different between the years based on annual (or growing season) sums. In the new analysis based on the daily values you now say that the difference is around 14% due to the different calculations. I find that contradicting and you should make clear why you do so and differentiate with which analysis and why you want to address shorter-term carbon dynamics or carbon sink functions of your ecosystem.
- I also want to say that I do not understand the logic behind using the daily values here but I might miss something.

**Response:** We would like to thank the reviewer for bringing this apparent contradiction to our attention! Indeed, the seasonally accumulated values of GPP for 2017 and 2018 were similar, whereas the resistance index calculated from daily GPP was 0.85, meaning a 15% decline. The discrepancy comes from the difference in temporal scales. Seasonal sums smooth out short-term variability: in 2018, higher GPP during the early growing season partly compensated for suppression in mid- to late summer (Figure 3), resulting in comparable seasonal totals. In contrast, the resistance index, calculated from the bootstrapped averages of daily values, captures these episodic declines more accurately, reflecting the stronger suppression of photosynthesis during the peak drought period.

Then, the bootstrapping approach is likely shifting around the day of year between the years randomly to explore what you call uncertainty here. What kind of uncertainty would that be? Accounting for different phenological development between different years.

**Response:** Since the variability of the daily ecosystem parameters can be rather high, the averages used in indices calculations could be affected by single outstanding days. To avoid this, we chose to use the bootstrapping approach.

I believe that if you open the pandoras box of uncertainty here you would rather start a step earlier and explore the uncertainties related to the eddy covariance method itself, more importantly the flux partitioning and finally your ML-based gap-filling approach. Note that I do not think you should do so for the manuscript but the bootstrapping approach based on daily values gives the reader a false impression that the uncertainties were understood while the real source of uncertainty lies elsewhere. From experience, I would go as far as speculate that a full analysis of the uncertainties related to flux partitioning might even result in the not being able (in a statistical sense) to differentiate between the individual years.

**Response:** We agree that the term "uncertainty" may have been misleading in this context. While the reported confidence intervals reflect the variability of the indices derived from bootstrapping, they, of course, do not represent the full uncertainty of the flux measurements themselves. We have rephrased the corresponding section in the Methods (lines 252-254) to avoid potential misinterpretation.

- The interpretation of the outcome is relatively brief and, in my opinion, does not go beyond what you already discussed effectively in your previous analysis. Consider better explaining what added benefit the analysis has and what arguments which have not been presented before can be drawn from it.
- Then, from l.594 you discuss, again this being an example for the general issue, the reduction in GPP which you previously have shown to be not significant based on the values in Table 1 (e.g., l.302). I would make sure to remove this contradiction and if there is some added value of these two different calculations discuss their importance and what added benefit they bring to your analysis.

**Response: We hope that the added explanation above cleared the discrepancy**

1.543 and Figure A1: Very creative approach to estimate the energy balance closure in the absence of data from a net radiometer and ground heat flux plates. Neglecting the longwave term will overestimate the true net radiation, as net longwave radiation is usually negative on an annual scale. This means that if you had included that term, your energy balance closure would actually appear better than shown in Fig. A1. From the turbulent flux perspective, this might therefore less problematic as interpreted. Regarding the approximation of G=0.05\*Rn, this relationship is probably more complex. I quickly checked that assumption for one year at an evergreen needleleaf forest site, where I just had the data at hand and found that 0.05\*Rn slightly overestimates the actual G with a linear regression suggestion 0.03\*Rn - 5 (with units being Wm-2, relevant for the intercept). Obviously, that's a very different site with different soil and moisture regime but one could argue that this is close enough in the broader picture. My recommendation would be that you briefly mention what you expect from neglecting the longwave term in Rn and leave the simple G model as it is or briefly say what errors you expect from it. Overall, in my opinion, you miss a chance here to say that your closure is actually better than you can show based on your limited instrumentation based on the neglection of the longwave term.

**Response:** Thank you for the positive and constructive feedback! We added the mention of the missing longwave radiation and the limitation of the simplified G calculation approach to the corresponding appendix text

l.544, l.551: Avoid statements as "sufficient GPP" and "adequate water supply". There are probably more of these kind of wordings throughout the manuscript. I recommend to double check carefully.

**Response**: We carefully reviewed the manuscript and revised instances of vague wording that we could identify.

*l.563: in brackets: I suggest removing the brackets and adding "representing" [the maximum stomatal aperture]*

**Response:** We agree. Rephrased, see line

l.648: I'm not sure here, but could it also be that a large amount of volatile carbon was available after ER was so low in 2018? You do indirectly suggest that in the sentence starting from line 653 anyways but not implicitly for your study.

**Response:** We agree that this is a plausible explanation. However, without direct measurements of heterotrophic respiration, we cannot confirm it, so it remains speculative. We added a sentence in the Discussion to clarify this possibility (lines 662-664)

*l.687: "High EWUE and reduced Gc..." That sentence seems to lack something. Could go: In consequence, high EWUE and reduced Gc....*

**Response:** High EWUE and reduced Gc were the indicators of stomatal regulation, rather than the consequences of declined GPP and ET

l.696: Completely optional and just my own opinion: I obviously haven't been to the forest but from my interpretation from all of your results I wouldn't think that this 2018 drought would cause a delayed tree mortality. While this is a more general statement and you say so, I find this final sentence a bit underwhelming given the results you produced. You already mention the call for continued long-term monitoring at the end of the discussion and your data just consists of 3 years before the station was moved. Maybe you could think of a better fitting last sentence suitable for your specific manuscript.

**Response:** We rephrased the final sentence: see lines 705-708